# Mixture Prototype Flow Matching for Open-Set Supervised Anomaly Detection

**Fuyun Wang**[1] **Yuanzhi Wang**[1] **Xu Guo**[2] **Sujia Huang**[1] **Tong Zhang**[1] **Dan Wang**[3] **Hui Yan**[1] **Xin Liu**[4] **Zhen Cui**[2]

## Abstract

Open-set supervised anomaly detection (OSAD) aims to identify unseen anomalies using limited anomalous supervision. However, existing prototype-based methods typically model normal data via a unimodal Gaussian prior, failing to capture inherent multi-modality and resulting in blurred decision boundaries. To address this, we propose Mixture Prototype Flow Matching (MPFM), a framework that learns a continuous transformation from normal feature distributions to a structured Gaussian mixture prototype space. Departing from traditional flow-based approaches that rely on a single velocity vector, MPFM explicitly models the velocity field as a Gaussian mixture prior where each component corresponds to a distinct normal class. This design facilitates mode-aware and semantically coherent distribution transport. Furthermore, we introduce a Mutual Information Maximization Regularizer (MIMR) to prevent prototype collapse and maximize normal-anomaly separability. Extensive experiments demonstrate that MPFM achieves state-of-the-art performance across diverse benchmarks under both single- and multi-anomaly settings. Code is available at https://github.com/fuyunwang/MPFM-OSAD.

## 1. Introduction

Anomaly detection (AD) plays a pivotal role in identifying deviations from normal patterns, underpinning critical applications such as industrial inspection and medical imaging (Yao et al., 2024). While recent advances in Unsupervised AD (Lu et al., 2023; Liu et al., 2024) and few-shot

[1]Nanjing University of Science and Technology, Nanjing, China [2]Beijing Normal University, Beijing, China [3]China Academy of Space Technology, Beijing, China [4]Nanjing SeetaCloud Technology, Nanjing, China. Correspondence to: Dan Wang <wang-dan_ict_hit@163.com>, Zhen Cui <zhen.cui@bnu.edu.cn>.

*Proceedings of the 43rd International Conference on Machine Learning*, Seoul, South Korea. PMLR 306, 2026. Copyright 2026 by the author(s).

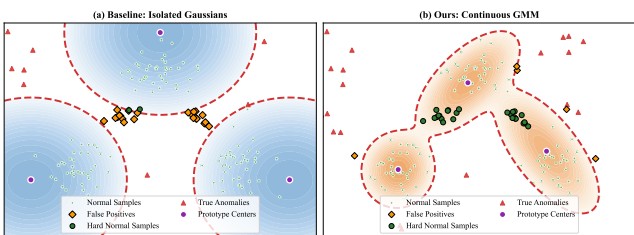

*Figure 1.* (a) Existing method assumes a unimodal normal distribution, overlooking intrinsic multi-modality and causing false positives. (b) Our method learns a continuous multi-modal prototype space via flow, capturing intra-class diversity, concentrating normal density, and enlarging the margin to true anomalies.

AD (FSAD) (Pang et al., 2021; Hu et al., 2024) focus on modeling normal distributions to flag irregularities, they often overlook the valuable prior information embedded in observed anomalies, resulting in imprecise decision boundaries. Conversely, Supervised AD (SAD) (Baitieva et al., 2024) explicitly incorporates known anomalies for guidance; however, its reliance on seen classes risks overfitting, leading to poor generalization across unseen anomaly types. To bridge this gap, Open-set Supervised AD (OSAD) (Ding et al., 2022; Wang et al., 2025a) aims to leverage limited anomalous examples with image-level supervision, ensuring robust detection of both known and previously unseen anomalies.

Existing OSAD methods can be grouped into three categories. The first employs data augmentation and outlier exposure (e.g., DRA (Ding et al., 2022)), which increases anomaly coverage by disentangling seen, pseudo, and residual factors, while other methods inject synthetic defects or external outliers to refine decision boundaries. The second category relies on heterogeneous simulation, as exemplified by AHL (Zhu et al., 2024), to enhance generalization using diverse simulated anomalies. The third leverages prototype learning and generative dynamics, constructing a structured prototype space where normal samples are drawn toward prototype centers and anomalies are repelled, often via diffusion or flow-based mechanisms. However, these methods face notable limitations: augmentation and exposure often cover only a limited anomaly spectrum and retain in-distribution bias, while simulation-based approaches struggle to capture the scale and structural diversity of

real-world anomalies. A recent prototype-based OSAD approach, DPDL (Wang et al., 2025a), models normal sample prototypes with a set of simple Gaussian distributions and identifies anomalies by constructing bridging flows that shift normal samples toward these prototypes. However, by assuming normal data is uni-modal and limited to observed categories, this method cannot represent the multi-modal complexity of real-world industrial data. As shown in Fig. 1 (a), a uni-modal Gaussian prior oversimplifies the sub-patterns within the normal sample distribution and fails to capture rare normal instances, leading to their misclassification as anomalies (i.e., false positives) and blurring the decision boundary of the normal samples. This raises a critical question: *How can we more precisely model the distribution of normal samples to establish a compact, comprehensive decision boundary that remains robust against unseen anomalies?*

To address these challenges, we propose Mixture Prototype Flow Matching (MPFM), a novel framework integrating Gaussian mixture modeling with continuous flow matching to explicitly capture the multi-modal nature of normal data. Departing from conventional uni-modal prior paradigms, MPFM characterizes normality within a structured Gaussian mixture prototype space, where each component represents a semantically distinct normal mode. Technically, we leverage flow matching to learn a mode-aware velocity field that transports features into this mixture prior, enabling fine-grained and mode-aligned distribution transformation. This yields a more expressive characterization of normality, enhancing both anomaly discrimination and generalization. Furthermore, to mitigate prototype collapse and ensure open-set separability, we introduce a Mutual Information Maximization Regularizer (MIMR), which enforces confident prototype assignments and margin-based separation from anomalies. Our contributions are outlined as follows:

- We propose Mixture Prototype Flow Matching (MPFM) to integrate Gaussian mixture priors with flow matching to capture multi-modal normality and produce sharper decision boundaries.

- We introduce a Mutual Information Maximization Regularizer (MIMR) to prevent mode collapse and enhance feature discriminability.

- Extensive experiments demonstrate that MPFM achieves state-of-the-art performance across diverse benchmarks.

## 2. Related Work

### 2.1. Open-set Supervised Anomaly Detection

Open-set supervised anomaly detection generalizes from limited known anomalies to unseen types, leveraging anoma-

lous priors to reduce false alarms. Existing approaches follow two main paradigms. The first paradigm employs data augmentation and simulation. DRA (Ding et al., 2022) disentangles seen, pseudo, and residual anomalies to expand coverage, while AHL (Zhu et al., 2024) simulates heterogeneous distributions via collaborative training. However, these methods typically cover limited anomaly subspaces and may inherit training biases, constraining generalization to truly novel anomalies. The second paradigm adopts prototype- or generative-based approaches, constructing compact normal representations with transport dynamics. DPDL (Wang et al., 2025a) builds a prototype space using diffusion processes to attract normals and repel anomalies. Yet, its single-mode prototype fails to capture normal data multimodality, causing false positives on rare normal patterns and ambiguous boundaries near subtle anomalies. Addressing these limitations, our approach is guided by a Gaussian mixture prior for multi-modal normality, empowered by flow matching for stable alignment, and refined by a center-aware regularizer to bolster component separation.

### 2.2. Flow Matching

Flow matching provides a simulation-free framework for learning vector fields that transport data between complex and simple distributions (Lipman et al., 2022; Liu et al., 2022; Esser et al., 2024; Chen et al., 2025). Rectified Flow (Liu et al., 2022) substantially advanced the field by introducing straight probability paths between distributions, simplifying both training and sampling dynamics. Subsequent efforts further improved efficiency and generality: InstaFlow (Liu et al., 2023) achieved single-step generation via rectified flow distillation, while Stable Diffusion 3 (Esser et al., 2024) and Flux (Labs, 2024) emerged as state-of-the-art text-to-image models by building upon flow matching for robust and generalized synthesis (Wang et al., 2025b). These advances establish flow matching as a scalable and stable alternative to diffusion models, particularly suited for tasks requiring efficient distribution alignment—a property we leverage in this work to construct compact and discriminative normal feature distributions for anomaly detection.

## 3. Preliminaries

We adopt the standard additive–Gaussian perturbation view used by diffusion models and instantiate flow matching on top of it. Let the clean feature be $\mathbf{z}_0 \in \mathbb{R}^D$, Gaussian noise $\varepsilon \sim \mathcal{N}(\mathbf{0}, \mathbf{I})$ is injected along a time schedule $(\alpha_t, \sigma_t)$:

$$\mathbf{z}_t = \alpha_t \mathbf{z}_0 + \sigma_t \varepsilon. \qquad (1)$$

In the discrete-time setting, the process can be represented via the transition from $\mathbf{z}_{t-\Delta t}$ to $\mathbf{z}_t$ as:

$$p(\mathbf{z}_t|\mathbf{z}_{t-\Delta t}) = \mathcal{N}\left(\mathbf{z}_t; \frac{\alpha_t}{\alpha_{t-\Delta t}}\mathbf{z}_{t-\Delta t}, \beta_{t,\Delta t}I\right), \quad (2)$$

where $\beta_{t,\Delta t} := \sigma_t^2 - \frac{\alpha_t^2}{\alpha_{t-\Delta t}^2}\sigma_{t-\Delta t}^2$. During the reverse process, marginalizing $\mathbf{z}_0$ recovers the reverse kernel $p(\mathbf{z}_{t-\Delta t}|\mathbf{z}_t) = \int p(\mathbf{z}_{t-\Delta t}|\mathbf{z}_t,\mathbf{z}_0)p(\mathbf{z}_0|\mathbf{z}_t)\,\mathrm{d}\mathbf{z}_0$:

$$p(\mathbf{z}_{t-\Delta t} \mid \mathbf{z}_t, \mathbf{z}_0) = \mathcal{N}\big(\mathbf{z}_{t-\Delta t}; c_1\mathbf{z}_t + c_2\mathbf{z}_0, c_3\mathbf{I}\big), \quad (3)$$

where $c_1 = \frac{\alpha_t \sigma_{t-\Delta t}^2}{\alpha_{t-\Delta t}\sigma_t^2}, c_2 = \frac{\alpha_{t-\Delta t}\beta_{t,\Delta t}}{\sigma_t^2}, c_3 = \frac{\sigma_{t-\Delta t}^2 \beta_{t,\Delta t}}{\sigma_t^2}$.

Flow Matching (FM) trains a neural network to directly predict the ODE velocity field $\frac{\mathrm{d}\mathbf{z}_t}{\mathrm{d}t}$. By adopting a linear noise schedule defined as $T = 1, \alpha_t = 1 - t$ and $\sigma_t = t$, the random velocity variable is defined as:

$$\mathbf{u} = \frac{\mathbf{z}_t - \mathbf{z}_{t-\Delta t}}{\Delta_t}, \quad (4)$$

During training, FM learns a time-dependent vector field by fitting a parametric conditional $q_\theta(\mathbf{u} \mid \mathbf{z}_t)$ to the ground truth $p(\mathbf{u} \mid \mathbf{z}_t)$ in Kullback-Leibler Divergence (KL):

$$\mathcal{L}_{\mathrm{FM}} = \mathbb{E}_{t,\mathbf{z}_0,\mathbf{z}_t}\big[D_{\mathrm{KL}}\big(p(\mathbf{u} \mid \mathbf{z}_t) \,\|\, q_\theta(\mathbf{u} \mid \mathbf{z}_t)\big)\big]$$
$$\equiv \mathbb{E}_{t,\mathbf{z}_0,\mathbf{z}_t}\big[-\log q_\theta(\mathbf{u} \mid \mathbf{z}_t)\big]. \quad (5)$$

Technically, we obtain $\mathbf{z}_t$ by sampling $\mathbf{z}_0$, drawing $\varepsilon$, and computing $\mathbf{u}$ via Eqn. (1). Minimizing Eqn. (5) is a supervised regression of velocities conditioned on $\mathbf{z}_t$:

$$\mathcal{L}_{\mathrm{FM}} = \mathbb{E}_{t,\mathbf{z}_0,\mathbf{z}_t}\big[\tfrac{1}{2}\|\mathbf{u} - \boldsymbol{\mu_\theta}(\mathbf{z}_t)\|_2^2\big], \quad (6)$$

where $\boldsymbol{\mu_\theta}(\mathbf{z}_t)$ denotes a flow velocity neural network with learnable parameters $\theta$.

## 4. Method

### 4.1. Problem Definition

Let $\mathcal{Z}_{\mathrm{tr}} = \{(\mathbf{z}_i, y_i)\}$ be a weakly-supervised training set with $\mathbf{z}_i$ denoting an image and $y_i \in \{0,1\}$ denotes image-level labels, where 0 indicates normal and 1 indicates anomaly. The training set typically divides as $\mathcal{Z}_{\mathrm{tr}} = \mathcal{Z}_{\mathrm{tr}}^{\mathrm{n}} \cup \mathcal{Z}_{\mathrm{tr}}^{\mathrm{a}}$, with normal subset $|\mathcal{Z}_{\mathrm{tr}}^{\mathrm{n}}| = N$, anomaly subset $|\mathcal{Z}_{\mathrm{tr}}^{\mathrm{a}}| = M$, and typically $N \gg M$. Given a test set $\mathcal{Z}_{\mathrm{te}}$, the goal is to predict whether $\mathbf{z} \in \mathcal{Z}_{\mathrm{te}}$ is normal or anomaly. Under OSAD, the anomaly distribution at test time differs significantly from that during training, i.e., $p_{\mathrm{te}}^{\mathrm{a}}(\mathbf{z}) \neq p_{\mathrm{tr}}^{\mathrm{a}}(\mathbf{z})$.

OSAD requires learning compact and discriminative representations of normality. However, methods based on discrete multi-Gaussian prototypes face a fundamental limitation: the lack of structured relationships among components fragments the prototype space and disrupts semantic continuity in the data manifold. While typically formulated as learning a set of independent Gaussians $\mathcal{N}(\boldsymbol{\mu}_k, \sigma_k^2\mathbf{I})$, this paradigm cannot establish semantic connections between modes, ultimately limiting its capacity to model complex normal distributions.

---

**Algorithm 1** MPFM: Mixture Prototype Flow Matching

---

**Require:** Training sets $\mathcal{Z}_{\mathrm{tr}}^{\mathrm{n}}$, $\mathcal{Z}_{\mathrm{tr}}^{\mathrm{a}}$; test set $\mathcal{Z}_{\mathrm{te}}$
**Ensure:** Anomaly scores $\{S(\mathbf{z})\}$ for $\mathbf{z} \in \mathcal{Z}_{\mathrm{te}}$
1: **Hyperparameters:** GMM components $K$, MIMR weight $\lambda$, Learning rate $\eta$
2: Initialize $f, \psi$ with $\theta$, scoring modules $\{\theta_{\mathrm{a}}, \theta_{\mathrm{n}}, \theta_{\mathrm{r}}, \theta_{\mathrm{g}}\}$
3: Extract features: $\mathcal{F}_{\mathrm{tr}}^{\mathrm{n}} \leftarrow f(\mathcal{Z}_{\mathrm{tr}}^{\mathrm{n}})$, $\mathcal{F}_{\mathrm{tr}}^{\mathrm{a}} \leftarrow f(\mathcal{Z}_{\mathrm{tr}}^{\mathrm{a}})$
4: Initialize GMM prototype via K-means++ on $\mathcal{F}_{\mathrm{tr}}^{\mathrm{n}}$:
5: $\quad \{\boldsymbol{\mu}_k, \pi_k\}_{k=1}^K \leftarrow$ K-means++$(\mathcal{F}_{\mathrm{tr}}^{\mathrm{n}})$
6: $\quad s^2 \leftarrow \frac{1}{dN}\sum_{k=1}^K \sum_{i \in C_k} \|\mathbf{z}_0^{\mathrm{n},i} - \boldsymbol{\mu}_k\|_2^2$
7: **while** not converged **do**
8: $\quad$ Sample batch: $\{\mathbf{z}_0^{\mathrm{n},i}\} \sim \mathcal{F}_{\mathrm{tr}}^{\mathrm{n}}$, $\{\mathbf{z}_0^{\mathrm{a},j}\} \sim \mathcal{F}_{\mathrm{tr}}^{\mathrm{a}}$
9: $\quad$ Sample timestep: $t \sim \mathcal{U}(0, T)$
10: $\quad$ Compute interpolated features and velocities:
11: $\quad\quad \mathbf{z}_t^{\mathrm{n},i} \leftarrow (1-t)\mathbf{z}_0^{\mathrm{n},i} + t\mathbf{z}_T^{\mathrm{n},i}$, $\mathbf{u}^{\mathrm{n},i} \leftarrow \frac{\mathbf{z}_T^{\mathrm{n},i} - \mathbf{z}_0^{\mathrm{n},i}}{t}$
12: $\quad\quad \mathbf{z}_t^{\mathrm{a},j} \leftarrow (1-t)\mathbf{z}_0^{\mathrm{a},j} + t\mathbf{z}_T^{\mathrm{a},j}$, $\mathbf{u}^{\mathrm{a},j} \leftarrow \frac{\mathbf{z}_T^{\mathrm{a},j} - \mathbf{z}_0^{\mathrm{a},j}}{t}$
13: $\quad$ Compute flow matching losses:
14: $\quad\quad \mathcal{L}_{\mathrm{flow}}^{\mathrm{n}} \leftarrow \mathbb{E}[-\log q_\theta(\mathbf{u}^{\mathrm{n},i}|\mathbf{z}_t^{\mathrm{n},i})]$
15: $\quad\quad \mathcal{L}_{\mathrm{flow}}^{\mathrm{a}} \leftarrow \mathbb{E}[\log q_\theta(\mathbf{u}^{\mathrm{a},j}|\mathbf{z}_t^{\mathrm{a},j})]$
16: $\quad$ Compute MIMR regularization:
17: $\quad\quad \mathcal{L}_{\mathrm{mim}} \leftarrow \mathbb{E}[\sum_k p(c = k|\psi(\mathbf{z}_0^{\mathrm{n},i})) \log p(c = k|\psi(\mathbf{z}_0^{\mathrm{n},i}))] - \sum_k \pi_k \log \pi_k$
18: $\quad$ Compute: $\mathcal{L}_{\mathrm{score}} \leftarrow \sum_{\mathrm{m} \in \{\mathrm{a},\mathrm{n},\mathrm{r},\mathrm{g}\}} \mathcal{L}_{\mathrm{M_m}}$
19: $\quad$ Total loss: $\mathcal{L} \leftarrow \mathcal{L}_{\mathrm{flow}}^{\mathrm{n}} + \mathcal{L}_{\mathrm{flow}}^{\mathrm{a}} + \lambda\mathcal{L}_{\mathrm{mim}} + \mathcal{L}_{\mathrm{score}}$
20: $\quad$ Update $\Theta \leftarrow \{\theta, \theta_{\mathrm{a}}, \theta_{\mathrm{n}}, \theta_{\mathrm{r}}, \theta_{\mathrm{g}}\}$: $\Theta \leftarrow \Theta - \eta \cdot \nabla_\Theta \mathcal{L}$
21: **end while**
22: **Inference:** For each $\mathbf{z} \in \mathcal{Z}_{\mathrm{te}}$:
23: $\quad S(\mathbf{z}) \leftarrow S_{\mathrm{g}} + S_{\mathrm{a}} + S_{\mathrm{r}} - S_{\mathrm{n}}$
24: **return** $\{S(\mathbf{z})\}$

---

*Our core idea* is to overcome the limitation of discrete multi-Gaussian prototypes by learning a continuous and structured prototype space through Mixture Prototype Flow Learning (MPFL). Rather than depending on a collection of isolated distributions, we introduce a generative bridge in the form of a flow model $\psi$, which seamlessly maps the complex feature distribution of normal data onto a Gaussian Mixture Model (GMM) prior. This formulation effectively captures the multi-modal nature and smooth transitions inherent in the normal data manifold, yielding a more coherent and expressive density estimator that mitigates false positive anomalies. Formally, we jointly optimize the flow mapping $\psi$ and the parameters of the Gaussian mixture prototype $\mathcal{P}_{\mathrm{GM}}$ under the following objective:

$$\min_{\psi, \mathcal{P}_{\mathrm{GM}}, f} D_{\mathrm{flow}}(\psi(\mathcal{F}_{\mathrm{tr}}^{\mathrm{n}}), \psi(\mathcal{F}_{\mathrm{tr}}^{\mathrm{a}}), \mathcal{P}_{\mathrm{GM}}) + \lambda D_{\mathrm{mim}}(\mathcal{F}_{\mathrm{tr}}^{\mathrm{n}}), \quad (7)$$

$$\text{s.t., } \psi : P(\mathcal{F}) \xrightarrow{\text{flow}} \mathcal{P}_{\mathrm{GM}}, \quad (8)$$

$$f : \mathcal{Z} \xrightarrow{\text{feature}} \mathcal{F}, \quad (9)$$

where $\mathcal{P}_{\mathrm{GM}}$ denotes the Gaussian mixture prototype to be learned, $\psi$ is the flow function that performs probability

distribution transport, $\mathcal{F}_{\text{tr}}^{\text{n}}$ and $\mathcal{F}_{\text{tr}}^{\text{a}}$ denote the feature space of normal and anomalous samples respectively, $D_{\text{flow}}$ represents the discriminative objective in the prototype space, $f$ is the feature extractor, and $D_{\text{mim}}$ serves as a feature-space regularizer. In this formulation, we refer to the learning of $\psi$ and $\mathcal{P}_{\text{GM}}$ as MPFL. Concurrently, the objective $D_{\text{mim}}(\mathcal{F}_{\text{tr}}^{\text{norm}})$ is referred to as Mutual Information Maximization Regularizer (MIMR), which encourages confident yet balanced prototype usage and reinforces normal–anomalous separation, thereby mitigating prototype collapse and enhancing overall discriminability. The complete algorithmic procedure is summarized in Algorithm 1.

### 4.2. Mixture Prototype Flow Learning

Our Mixture Prototype Flow Learning (MPFL) is performed in the feature space. To extract intermediate image features, we employ a classic backbone network such as ResNet-18. Formally, the feature extraction process can be described by a function $f : \mathcal{Z} \rightarrow \mathcal{F}$, which maps an image $z$ to an intermediate feature representation $\mathbf{z} = f(z) \in \mathbb{R}^d$, where $\mathbb{R}^d$ denotes a $d$-dimensional real space[1]. We designate the intermediate feature sets of normal and anomaly samples as $\mathcal{F}_{\text{tr}}^{\text{n}} = \{\mathbf{z}_0^{\text{n},i} | i = 1, \cdots, N\}$ and $\mathcal{F}_{\text{tr}}^{\text{a}} = \{\mathbf{z}_0^{\text{a},j} | j = 1, \cdots, M\}$, respectively. Throughout the subsequent work, we adopt a unified notation: subscripts denote timesteps (e.g., $\mathbf{z}_0$) and superscripts indicate both sample types and indices (e.g., $\mathbf{z}_0^{\text{n},i}$ for the $i$-th normal sample at timestep 0, $\mathbf{z}_0^{\text{a},j}$ for the $j$-th anomalous sample at timestep 0).

Existing prototype-based OSAD methods rely on a simple unimodal Gaussian prior, which significantly underestimates the inherent multimodality of normal data. To mitigate this limitation, MPFL learns continuous flow transformations to map the features of normal samples to a structured Gaussian mixture prior, effectively capturing the complex and multi-modal characteristics of normal data. Specifically, owing to the scarcity of anomalous samples and the abundance and diversity of normal data, we consider learning a continuous distribution transformation from the source distribution $p_{\text{source}} = P(\mathcal{F}_{\text{tr}}^{\text{n}})$ of normal samples to the target Gaussian mixture prototype distribution $p_{\text{target}} = P(\mathcal{P}_{\text{GM}})$, which is defined as:

$$p_1(\psi(\mathbf{z}_0^{\text{n},i})) = \sum_{k=1}^{K} \pi_k \mathcal{N}(\psi(\mathbf{z}_0^{\text{n},i}); \boldsymbol{\mu}_k, s^2\mathbf{I}), \quad (10)$$

where $\psi$ denotes the flow function and $\psi(\mathbf{z}_0^{\text{n},i}) \in \mathbb{R}^d$ denotes the latent features. $K$ is the number of mixture components, $\pi_k$ is the weight of the $k$-th component satisfying $\pi_k > 0$ and $\sum_{k=1}^{K} \pi_k = 1$, $\boldsymbol{\mu}_k \in \mathbb{R}^d$ is mean vector, and $s \in \mathbb{R}^+$ is a shared standard deviation to maintain numerical

---

[1]$\mathbf{z}$ is obtained by flattening the spatial dimensions of the convolutional feature maps, transforming the 3D tensor into a 1D representation.

stability.

Traditional flow-matching methods typically regress a single velocity vector, which limits their ability to capture the complex multimodal characteristics of normal data. To mitigate this limitation, we directly model the velocity field $p(\mathbf{u}|\mathbf{z}_t^{\text{n},i})$ as a GMM:

$$q_\theta(\mathbf{u}|\mathbf{z}_t^{\text{n},i}) = \sum_{k=1}^{K} \pi_k(\mathbf{z}_t^{\text{n},i}; \theta)\mathcal{N}(\mathbf{u}; \boldsymbol{\mu}_k(\mathbf{z}_t^{\text{n},i}; \theta), s^2\mathbf{I}), \quad (11)$$

where $\mathbf{u} \in \mathbb{R}^d$ denotes the velocity vector, and $\mathbf{z}_t^{\text{n},i}$ represents the feature representation at timestep $t$ obtained by Eqn. (4). Building on Eqn. (11), we directly regress the conditional velocity distribution with a Gaussian mixture. Training minimizes the following negative log-likelihood:

$$\mathcal{L}_{\text{NLL}} = \mathbb{E}_{\mathbf{u} \sim p(\mathbf{u}|\mathbf{z}_t^{\text{n},i})}$$
$$\left[ -\log \sum_{k=1}^{K} \pi_k(\mathbf{z}_t^{\text{n},i}; \theta) \, \mathcal{N}\left(\mathbf{u}; \boldsymbol{\mu}_k(\mathbf{z}_t^{\text{n},i}; \theta), s^2\mathbf{I}\right) \right]. \quad (12)$$

We introduce an explicit time parameterization that enables closed-form reverse-time sampling while preserving the Gaussian mixture structure. Under a standard linear noise schedule, the one-step transition admits an approximate reparameterization, yielding a conditional endpoint distribution consistent with the velocity field:

$$q_\theta(\mathbf{z}_0^{\text{n},i}|\mathbf{z}_t^{\text{n},i}) = \sum_{k=1}^{K} \pi_k(\mathbf{z}_0^{\text{n},i}; \boldsymbol{\mu}_{z_k}, s_z^2\mathbf{I}),$$
$$\text{s.t.,} \quad \boldsymbol{\mu}_{z_k} = \mathbf{z}_t - \sigma_t\boldsymbol{\mu}_k, \quad s_z = \sigma_t s, \quad (13)$$

By the conditional–marginal closure of linear–Gaussian models, the one-step reverse transition that preserves the mixture structure is defined as follows:

$$q_\theta(\mathbf{z}_{t-\Delta t}^{\text{n},i} \mid \mathbf{z}_t^{\text{n},i}) = \sum_{k=1}^{K} \pi_k(\mathbf{z}_t^{\text{n},i}; \theta)$$
$$\mathcal{N}\left(\mathbf{z}_{t-\Delta t}; c_1\mathbf{z}_t^{\text{n},i} + c_2\boldsymbol{\mu}_{z_k}, (c_3 + c_2^2 s_z^2)\mathbf{I}\right), \quad (14)$$

where the coefficients are fully determined by the noise schedule: $c_1 = \frac{\sigma_{t-\Delta t}^2}{\sigma_t^2}\frac{\alpha_t}{\alpha_{t-\Delta t}}, c_2 = \frac{\beta_{t,\Delta t}}{\sigma_t^2}\alpha_{t-\Delta t}, c_3 = \frac{\beta_{t,\Delta t}}{\sigma_t^2}\sigma_{t-\Delta t}^2$. The closed-form transition enables step-wise sampling at inference without numerical integration, while preserving mixture closure throughout the reverse trajectory, yielding numerically stable inference.

However, the interdependence between the Gaussian mixture parameters and the flow transformation network presents a significant optimization challenge. To address this limitation, we introduce a data-driven initialization strategy for the target Gaussian mixture prototype distribution

$P(\mathcal{P}_{\mathrm{GM}})$. Specifically, we perform K-means++ clustering on $\mathcal{F}_{\mathrm{tr}}^{\mathrm{n}}$ to partition it into $K$ clusters $\{C_k\}_{k=1}^{K}$, where $C_k$ denotes the set of sample indices belonging to the $k$-th cluster, minimizing the within-cluster sum of squares:

$$\min_{\{C_k, \boldsymbol{\mu}_k\}_{k=1}^{K}} \sum_{k=1}^{K} \sum_{i \in C_k} \|\mathbf{z}_0^{\mathrm{n},i} - \boldsymbol{\mu}_k\|_2^2. \qquad (15)$$

The resulting cluster centroids are used to initialize the mixture means $\boldsymbol{\mu}_k$. The mixture weights $\pi_k$ are initialized proportionally to cluster sizes as $\pi_k = |C_k|/N$, while the shared variance $s^2$ is set to the per-dimension average variance across all clusters:

$$s^2 = \frac{1}{dN} \sum_{k=1}^{K} \sum_{i \in C_k} \|\mathbf{z}_0^{\mathrm{n},i} - \boldsymbol{\mu}_k\|_2^2. \qquad (16)$$

We formulate the training objectives that enable effective learning of MPFL. For normal samples, we employ a negative log-likelihood loss to align the predicted velocity distribution with the ground truth dynamics:

$$\mathcal{L}_{\mathrm{flow}}^{\mathrm{n}} = \mathbb{E}_{t, \mathbf{z}_0^{\mathrm{n},i}, \mathbf{z}_t^{\mathrm{n},i}} \left[ -\log q_\theta(\mathbf{u}|\mathbf{z}_t^{\mathrm{n},i}) \right], \qquad (17)$$

where $t \sim \mathcal{U}(0, T)$, the true velocity $\mathbf{u} = \frac{\mathbf{z}_T^{\mathrm{n},i} - \mathbf{z}_0^{\mathrm{n},i}}{t}$ and the interpolated feature $\mathbf{z}_t^{\mathrm{n},i} = (1-t)\mathbf{z}_0^{\mathrm{n},i} + t\mathbf{z}_T^{\mathrm{n},i}$. This objective encourages the model to learn velocity distributions that faithfully capture the transformation dynamics of normal samples toward $P(\mathcal{P}_{\mathrm{GM}})$. For anomalous samples, we adopt an opposite strategy that pushes their distributions away from the normal prototype space:

$$\mathcal{L}_{\mathrm{flow}}^{\mathrm{a}} = \mathbb{E}_{t, \mathbf{z}_0^{\mathrm{a},i}, \mathbf{z}_t^{\mathrm{a},i}} \left[ \log q_\theta(\mathbf{u}|\mathbf{z}_t^{\mathrm{a},i}) \right]. \qquad (18)$$

where $\mathbf{z}_0^{\mathrm{a},i} \sim P(\mathcal{F}_{\mathrm{tr}}^{\mathrm{a}})$. The true velocity and the interpolated feature were computed in the same way as above.

### 4.3. Mutual Information Maximization Regularizer

Although MPFL captures the multi-modal distribution of normal data, the lack of explicit inter-prototype separation can induce prototype collapse, where multiple components converge to similar modes. This undermines discriminative power, making semantically distinct patterns indistinguishable in the latent space. To mitigate this challenge, we introduce a Mutual Information Maximization Regularizer (MIMR) that complements our flow-based Gaussian-mixture framework. The core insight is that optimal prototype learning should (i) encourage confident assignments of features to specific prototypes and (ii) maintain balanced utilization across components. Formally, let $c \in \{1, ..., K\}$ denote the prototype assignment variable, mutual information

$I(\psi(\mathbf{z}_0^{\mathrm{n},i}); c)$ measures the reduction in uncertainty about $c$ given $\psi(\mathbf{z}_0^{\mathrm{n},i})$, which is defined as:

$$I(\psi(\mathbf{z}_0^{\mathrm{n},i}); c) = H(c) - H(c|\psi(\mathbf{z}_0^{\mathrm{n},i})), \qquad (19)$$

where $H(c)$ is the marginal entropy of prototype assignments and $H(c|\psi(\mathbf{z}_0^{\mathrm{n},i}))$ is the conditional entropy.

Thanks to the Gaussian-mixture structure in Eqn. (10), both terms can be estimated without extra parameters. In particular, the posterior assignment follows directly from the prototype GMM:

$$p(c = k|\psi(\mathbf{z}_0^{\mathrm{n},i})) = \frac{\pi_k \mathcal{N}(\psi(\mathbf{z}_0^{\mathrm{n},i}); \boldsymbol{\mu}_k, s^2\mathbf{I})}{\sum_{j=1}^{K} \pi_j \mathcal{N}(\psi(\mathbf{z}_0^{\mathrm{n},i}); \boldsymbol{\mu}_j, s^2\mathbf{I})}. \qquad (20)$$

This posterior distribution enables efficient computation of the information-theoretic regularizer:

$$\mathcal{L}_{\mathrm{mim}} = \mathbb{E}_{\mathbf{z}_0^{\mathrm{n},i} \sim P(\mathcal{F}_{\mathrm{tr}}^{\mathrm{n}})}$$
$$\left[ \sum_{k=1}^{K} p(c = k|\psi(\mathbf{z}_0^{\mathrm{n},i})) \log p(c = k|\psi(\mathbf{z}_0^{\mathrm{n},i})) \right]$$
$$- \sum_{k=1}^{K} \pi_k \log \pi_k, \qquad (21)$$

where the first term reduces conditional entropy (promoting confident assignments), and the second term increases marginal entropy (balancing component usage) via the mixture weights $\pi_k$. We evaluate $\mathcal{L}_{\mathrm{mim}}$ only on normal samples to prevent pulling anomalies toward the normal prototypes.

### 4.4. Anomaly Score Prediction

Based on the MPFM framework, we design four complementary modules for estimating anomaly scores from different perspectives. For the feature map $\mathbf{F} \in \mathbb{R}^{H' \times W' \times C}$[2] extracted from input images, we generate pixel-wise feature vectors $\mathcal{V} = \{\mathbf{v}_i\}_{i=1}^{H' \times W'}$ where each $\mathbf{v}_i \in \mathbb{R}^C$ represents the feature of a local image patch. We design the global anomaly scoring module $\mathrm{M_g}$ that computes the negative log-likelihood of transformed features under the Gaussian mixture prototype distribution:

$$\mathcal{L}_{\mathrm{M_g}} = \mathcal{L}_{\mathrm{binary}}(-\log \sum_{k=1}^{K} \pi_k \mathcal{N}(\psi(\mathbf{z}); \boldsymbol{\mu}_k, s^2\mathbf{I}), y), \quad (22)$$

where $\mathcal{L}_{\mathrm{binary}}$ refers to a binary classification loss function. The local anomaly scoring module $\mathrm{M_a}$ focuses on the most salient anomalous regions by aggregating top-$O$ pixel-level anomaly scores:

$$\mathcal{L}_{\mathrm{M_a}} = \mathcal{L}_{\mathrm{binary}}(\frac{1}{O} \sum \mathrm{Top}_O\{S_{\mathrm{a}}(\mathbf{v}_i; \theta_{\mathrm{a}})\}, y), \qquad (23)$$

---

[2]The spatial feature map $\mathbf{F}$ is taken from the convolutional layers of $f$ by preserving the 3D tensor, while $\mathbf{z}$ is its flattened embedding.

The normal feature scoring module $M_n$ learns prototypical normal patterns through global feature pooling:

$$\mathcal{L}_{M_n} = \mathcal{L}_{\text{binary}}(S_n(\frac{1}{H' \times W'} \sum_{i=1}^{H' \times W'} \mathbf{v}_i; \theta_n), y) \quad (24)$$

The residual anomaly scoring module $M_r$ measures the deviation from the most probable prototype:

$$\mathcal{L}_{M_r} = \mathcal{L}_{\text{binary}}(S_r((\psi(\mathbf{z}) - \boldsymbol{\mu}_{c^*})/s; \theta_r), y), \quad (25)$$

where $c^* = \arg \max_c \mathcal{N}(\psi(\mathbf{z}); \boldsymbol{\mu}_c, s^2 \mathbf{I})$ denotes the index of the most probable prototype.

**Training.** During the training phase, the flow matching model, Gaussian mixture prototype model, and four anomaly scoring modules are jointly trained. To this end, we employ an objective function that encompasses three components as follows:

$$\mathcal{L} = \underbrace{\mathcal{L}_{M_a} + \mathcal{L}_{M_n} + \mathcal{L}_{M_r} + \mathcal{L}_{M_g}}_{\text{Anomaly scoring modules}} + \underbrace{\mathcal{L}_{\text{flow}}^n + \mathcal{L}_{\text{flow}}^a}_{\text{Flow matching}} + \lambda \mathcal{L}_{\text{mim}}.$$
$$(26)$$

where the coefficient $\lambda$ modulates the relative importance of the prototype regularization loss, and the learnable parameters include the flow model parameters $\theta$ and the scoring module parameters $\{\theta_a, \theta_n, \theta_r, \theta_g\}$.

**Inference.** During the test phase, we compute the anomaly score by adding the scores from $S_g$, $S_a$ and $S_r$, while subtracting the normal score obtained from $S_n$.

*In summary, we incorporate a shared standard deviation parameter $s$ for training stability and provide complete derivations for this and other formulations in the appendix.*

## 5. Experiment

### 5.1. Datasets and Evaluation Metric

**Datasets.** We evaluate MPFM on nine real-world anomaly detection benchmarks, covering six industrial defect datasets (MVTec AD (Bergmann et al., 2019), Optical (Wieler & Hahn, 2007), SDD (Tabernik et al., 2020), AITEX (Silvestre-Blanes et al., 2019), ELPV (Deitsch et al., 2019), Mastcam (Kerner et al., 2020)) and three medical datasets (Hyper-Kvasir (Borgli et al., 2020), Brain-MRI (Salehi et al., 2021), HeadCT (Salehi et al., 2021)). Following OSAD baselines (Ding et al., 2022; Zhu et al., 2024), we adopt two sampling protocols: a general protocol that randomly draws anomaly examples from all anomaly categories, and a hard protocol that samples from a single category to assess generalization to novel or unseen anomaly classes.

**Evaluation Metric.** Performance is assessed using Area Under the ROC Curve (AUC) across all methods and settings, with scores averaged over five independent runs.

### 5.2. Baselines

We evaluate MPFM against six baselines: SAOE (Markovitz et al., 2020; Tack et al., 2020; Li et al., 2021), MLEP (Liu et al., 2019), FLOS (Ross & Dollár, 2017), DRA (Ding et al., 2022), AHL (Zhu et al., 2024), and the prior DPDL (Wang et al., 2025a). Among these, MLEP, DRA, AHL, and DPDL are tailored for OSAD; SAOE is a supervised detector augmented with synthetic anomalies and anomaly exposure, whereas FLOS handles class imbalance via focal loss.

### 5.3. Implementation Details

The input image size is $448 \times 448 \times 3$. We set $O$ to 10% of the total scores in each score map. We optimize with AdamW (Loshchilov, 2017) using an initial learning rate of $2 \times 10^{-4}$ and weight decay of $1 \times 10^{-5}$. MPFM is trained on a single NVIDIA GeForce RTX 5090 for 50 epochs with 20 iterations per epoch. Following prior protocol(Ding et al., 2022), we evaluate under $M = 10$ and $M = 1$ anomalous-sample settings. To enhance robustness to unseen anomalies, we employ CutMix (Yun et al., 2019) to generate pseudo-anomaly samples as augmentations of known anomalies. GMM components $K$ is set to 32 and MIM weight $\lambda$ is set to 0.1 in our experiments.

### 5.4. Results under General Setting

Tab. 1 presents comprehensive AUC results under the general setting. MPFM achieves the best or tied-best mean AUC across all benchmarks, showing consistent advantages over the strongest baseline in each dataset. Under the one-shot setting, MPFM obtains the largest gains on BrainMRI, ELPV, Hyper-Kvasir and SDD, with improvements of 2.0, 1.7, 1.4, and 1.2 percentage points, respectively. These results demonstrate its robustness in data-scarce scenarios, especially for heterogeneous medical anomalies and subtle industrial defects. MPFM also improves AUC by 1.6 points on both Optical and AITEX, and by 1.4, 1.3, and 1.1 points on Mastcam, HeadCT, and MVTec AD, respectively, indicating stable generalization across diverse anomaly domains. Under the ten-shot setting, MPFM further shows notable gains on Mastcam, Hyper-Kvasir, ELPV, and AITEX, improving the strongest baselines by 3.4, 2.6, 2.5, and 1.7 percentage points, respectively.

### 5.5. Results under the Hard Setting

Tab. 2 reports the AUC results under the challenging hard setting. MPFM achieves the highest mean AUC across all datasets in both one-shot and ten-shot configurations, demonstrating strong robustness in more complex open-set scenarios. Under the ten-shot setting, MPFM obtains the most notable gains on Mastcam, AITEX, ELPV, and Carpet, surpassing the strongest baselines by 2.3, 1.8, 1.4, and

*Table 1.* AUC results (mean ± std) on nine real-world AD datasets under the general setting, with extended results in appendix. Best results are highlighted in **red** and second-best in **blue** (per column). Baseline state-of-the-art numbers are taken from the original papers.

| Dataset | FLOS | SAOE | MLEP | DRA | AHL | DPDL | MPFM (Ours) |
|---|---|---|---|---|---|---|---|
| | | | Ten Training Anomaly Examples | | | | |
| **MVTec AD** | 0.939±0.007 | 0.926±0.010 | 0.907±0.005 | 0.959±0.003 | 0.970±0.002 | **0.977**±0.002 | **0.982**±0.003 |
| **Optical** | 0.720±0.055 | 0.941±0.013 | 0.740±0.039 | 0.965±0.006 | 0.976±0.004 | **0.983**±0.005 | **0.992**±0.002 |
| **SDD** | 0.967±0.018 | 0.955±0.020 | 0.983±0.013 | 0.991±0.005 | 0.991±0.001 | **0.996**±0.001 | **0.999**±0.001 |
| **AITEX** | 0.841±0.049 | 0.874±0.024 | 0.867±0.037 | 0.893±0.017 | 0.925±0.013 | **0.975**±0.007 | **0.992**±0.004 |
| **ELPV** | 0.818±0.032 | 0.793±0.047 | 0.794±0.047 | 0.845±0.013 | 0.850±0.004 | **0.937**±0.003 | **0.962**±0.003 |
| **Mastcam** | 0.703±0.029 | 0.810±0.029 | 0.798±0.026 | 0.848±0.008 | 0.855±0.005 | **0.934**±0.010 | **0.968**±0.007 |
| **Hyper-Kvasir** | 0.773±0.029 | 0.666±0.050 | 0.600±0.069 | 0.834±0.004 | 0.880±0.003 | **0.939**±0.005 | **0.965**±0.004 |
| **BrainMRI** | 0.955±0.011 | 0.900±0.041 | 0.959±0.011 | 0.970±0.003 | **0.977**±0.001 | 0.969±0.005 | **0.984**±0.004 |
| **HeadCT** | 0.971±0.004 | 0.935±0.021 | 0.972±0.014 | 0.972±0.002 | **0.999**±0.003 | **0.981**±0.003 | **0.999**±0.001 |
| | | | One Training Anomaly Example | | | | |
| **MVTec AD** | 0.755±0.136 | 0.834±0.007 | 0.744±0.019 | 0.883±0.008 | 0.901±0.003 | **0.927**±0.002 | **0.938**±0.004 |
| **Optical** | 0.518±0.003 | 0.815±0.014 | 0.516±0.009 | 0.888±0.012 | 0.888±0.007 | **0.915**±0.002 | **0.931**±0.007 |
| **SDD** | 0.840±0.043 | 0.781±0.009 | 0.811±0.045 | 0.859±0.014 | 0.909±0.001 | **0.917**±0.003 | **0.929**±0.006 |
| **AITEX** | 0.538±0.073 | 0.675±0.094 | 0.564±0.055 | 0.692±0.124 | 0.734±0.008 | **0.838**±0.008 | **0.854**±0.023 |
| **ELPV** | 0.457±0.056 | 0.635±0.092 | 0.578±0.062 | 0.675±0.024 | 0.828±0.005 | **0.897**±0.002 | **0.914**±0.016 |
| **Mastcam** | 0.542±0.017 | 0.662±0.018 | 0.625±0.045 | 0.692±0.058 | 0.743±0.003 | **0.838**±0.011 | **0.852**±0.009 |
| **Hyper-Kvasir** | 0.668±0.004 | 0.498±0.100 | 0.445±0.040 | 0.690±0.017 | 0.768±0.015 | **0.821**±0.007 | **0.835**±0.005 |
| **BrainMRI** | 0.693±0.036 | 0.531±0.060 | 0.632±0.017 | 0.744±0.004 | 0.866±0.004 | **0.893**±0.004 | **0.913**±0.014 |
| **HeadCT** | 0.698±0.092 | 0.597±0.022 | 0.758±0.038 | 0.796±0.105 | 0.825±0.014 | **0.865**±0.005 | **0.878**±0.025 |

*Table 2.* AUC results (mean ± std) under the hard setting, with extended results in appendix. The best and second-best scores are highlighted in **red** and **blue**, respectively. Carpet and Metal_nut are subsets of MVTec AD. Following prior work (Ding et al., 2022; Zhu et al., 2024), we use the same datasets, excluding those with only a single anomaly class to align with the hard setting.

| Dataset | FLOS | SAOE | MLEP | DRA | AHL | DPDL | MPFM (Ours) |
|---|---|---|---|---|---|---|---|
| | | | Ten Training Anomaly Examples | | | | |
| **Carpet** (mean) | 0.761±0.012 | 0.762±0.073 | 0.751±0.023 | 0.935±0.013 | 0.949±0.002 | **0.956**±0.004 | **0.966**±0.003 |
| **Metal_nut** (mean) | 0.922±0.014 | 0.855±0.016 | 0.878±0.058 | 0.945±0.017 | 0.972±0.002 | **0.978**±0.002 | **0.983**±0.004 |
| **AITEX** (mean) | 0.635±0.043 | 0.724±0.032 | 0.626±0.041 | 0.733±0.009 | 0.747±0.002 | **0.798**±0.005 | **0.816**±0.006 |
| **ELPV** (mean) | 0.642±0.032 | 0.683±0.047 | 0.745±0.020 | 0.766±0.029 | 0.788±0.003 | **0.818**±0.003 | **0.832**±0.009 |
| **Mastcam** (mean) | 0.616±0.021 | 0.697±0.014 | 0.588±0.016 | 0.695±0.004 | 0.721±0.003 | **0.778**±0.007 | **0.801**±0.005 |
| **Hyper-Kvasir** (mean) | 0.786±0.021 | 0.698±0.012 | 0.571±0.014 | 0.844±0.009 | 0.854±0.004 | **0.864**±0.002 | **0.873**±0.017 |
| | | | One Training Anomaly Example | | | | |
| **Carpet** (mean) | 0.678±0.040 | 0.753±0.055 | 0.679±0.029 | 0.901±0.006 | 0.932±0.003 | **0.941**±0.006 | **0.949**±0.003 |
| **Metal_nut** (mean) | 0.855±0.024 | 0.816±0.029 | 0.825±0.023 | 0.932±0.017 | 0.939±0.004 | **0.944**±0.003 | **0.951**±0.004 |
| **AITEX** (mean) | 0.624±0.024 | 0.674±0.034 | 0.466±0.030 | 0.684±0.033 | 0.707±0.007 | **0.753**±0.005 | **0.772**±0.011 |
| **ELPV** (mean) | 0.691±0.008 | 0.614±0.048 | 0.566±0.111 | 0.703±0.022 | 0.740±0.003 | **0.762**±0.003 | **0.779**±0.008 |
| **Mastcam** (mean) | 0.524±0.013 | 0.689±0.037 | 0.541±0.007 | 0.667±0.012 | 0.673±0.010 | **0.733**±0.004 | **0.760**±0.013 |
| **Hyper-Kvasir** (mean) | 0.571±0.004 | 0.406±0.018 | 0.480±0.044 | 0.700±0.009 | 0.706±0.007 | **0.715**±0.004 | **0.739**±0.003 |

1.0 percentage points, respectively. These improvements suggest that Gaussian mixture prototypes can better characterize multi-modal normal patterns, while conditional flow matching helps preserve discriminative boundaries for diverse anomaly types. MPFM also improves the AUC by 0.9 points on Hyper-Kvasir and 0.5 point on Metal_nut, indicating stable performance even when the baseline results are already strong. Under the one-shot setting, MPFM maintains clear advantages, with gains of 2.7 points on Mastcam, 2.4 points on Hyper-Kvasir, 1.9 points on AITEX, 1.7 points on ELPV, 0.8 points on Carpet, and 0.7 points on Metal_nut.

## 5.6. Ablation Study

The ablation study in Fig. 2 systematically evaluates the contributions of core components in MPFM. Removing MPFL leads to substantial performance degradation across all datasets, particularly on complex-texture benchmarks like AITEX and Mastcam, underscoring its role in capturing multi-modal normality through Gaussian mixture prototypes. The MIMR removal results in moderate declines, especially in challenging settings, confirming its utility in preventing prototype collapse and enhancing feature discriminability. The synergistic combination of both components is evidenced by the maximal performance drop when ablating them jointly, highlighting their complementary functions in establishing compact normal boundaries and generalizing to unseen anomalies.

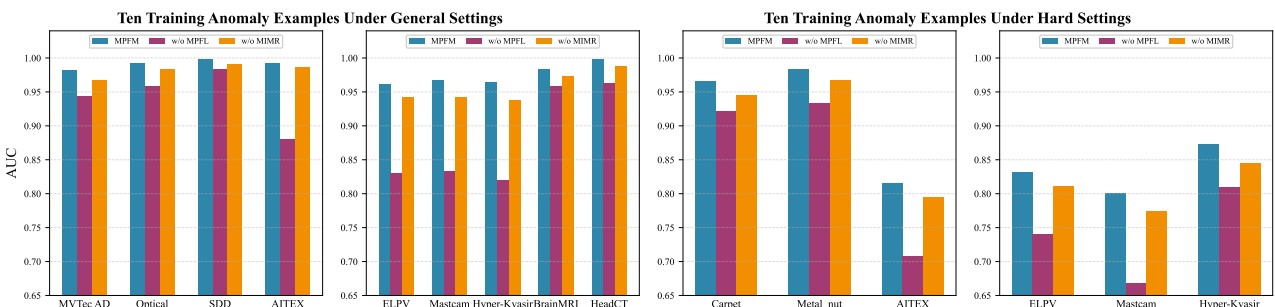

*Figure 2.* Ablation study for MPFL and MIMR under the general settings and hard settings.

*Table 3.* Ablation study for scoring modules $M_g$, $M_a$, $M_r$ and $M_n$.

| $M_g$ | $M_a$ | $M_r$ | $M_n$ | AITEX | Hyper-Kvasir | ELPV | Optical | Mastcam | HeadCT | BrainMRI |
|---|---|---|---|---|---|---|---|---|---|---|
| | | | | Ten Training Anomaly Examples Under General Settings | | | | | | |
| ✗ | ✓ | ✓ | ✓ | $0.945 \pm 0.009$ | $0.949 \pm 0.010$ | $0.946 \pm 0.006$ | $0.973 \pm 0.008$ | $0.948 \pm 0.014$ | $0.980 \pm 0.008$ | $0.962 \pm 0.007$ |
| ✓ | ✓ | ✗ | ✓ | $0.968 \pm 0.007$ | $0.959 \pm 0.005$ | $0.958 \pm 0.004$ | $0.982 \pm 0.005$ | $0.962 \pm 0.004$ | $0.987 \pm 0.006$ | $0.971 \pm 0.003$ |
| ✗ | ✓ | ✗ | ✓ | $0.932 \pm 0.012$ | $0.938 \pm 0.011$ | $0.929 \pm 0.015$ | $0.947 \pm 0.009$ | $0.937 \pm 0.013$ | $0.968 \pm 0.011$ | $0.934 \pm 0.012$ |
| ✓ | ✓ | ✓ | ✓ | $\mathbf{0.992} \pm 0.004$ | $\mathbf{0.965} \pm 0.004$ | $\mathbf{0.962} \pm 0.003$ | $\mathbf{0.992} \pm 0.002$ | $\mathbf{0.968} \pm 0.007$ | $\mathbf{0.999} \pm 0.001$ | $\mathbf{0.984} \pm 0.004$ |

As shown in Tab. 3, the scoring modules demonstrate distinct contributions to MPFM's performance. The most substantial performance degradation occurs when removing the global likelihood scorer $M_g$, underscoring its fundamental role in capturing normal patterns. The residual anomaly scorer $M_r$ also proves essential, with its removal leading to noticeable performance drops, particularly in complex detection scenarios. The most severe performance reduction emerges when both $M_g$ and $M_r$ are ablated simultaneously, highlighting their synergistic effect in maintaining robust anomaly discrimination across diverse settings.

*Table 4.* Parameter sensitivity analysis of GMM components $K$ and MIMR weight $\lambda$ on MVTec AD, AITEX, and Hyper-Kvasir under the General Setting.

| Param. | Value | MVTec AD | AITEX | Hyper-Kvasir |
|---|---|---|---|---|
| Components ($K$) | 8 | $0.975 \pm 0.009$ | $0.980 \pm 0.006$ | $0.954 \pm 0.017$ |
| | 16 | $0.979 \pm 0.008$ | $0.988 \pm 0.004$ | $0.960 \pm 0.010$ |
| | **32** | $\mathbf{0.982} \pm 0.003$ | $\mathbf{0.992} \pm 0.002$ | $\mathbf{0.965} \pm 0.004$ |
| | 64 | $0.978 \pm 0.012$ | $0.989 \pm 0.008$ | $0.958 \pm 0.016$ |
| MIMR Weight ($\lambda$) | 0.001 | $0.972 \pm 0.008$ | $0.987 \pm 0.010$ | $0.948 \pm 0.011$ |
| | 0.01 | $0.976 \pm 0.006$ | $0.989 \pm 0.005$ | $0.957 \pm 0.010$ |
| | **0.1** | $\mathbf{0.982} \pm 0.003$ | $\mathbf{0.992} \pm 0.002$ | $\mathbf{0.965} \pm 0.004$ |
| | 1.0 | $0.975 \pm 0.009$ | $0.981 \pm 0.007$ | $0.954 \pm 0.014$ |

### 5.7. Parameter Sensitivity Analysis

We analyze the sensitivity of the Gaussian components $K$ and the MIMR regularization weight $\lambda$. Tab. 4 reports the AUC on three representative datasets: MVTec AD (Object), AITEX (Texture), and Hyper-Kvasir (Medical).

**Impact of Component Number $K$.** Increasing the number of mixture components $K$ initially yields consistent performance gains, with optimal results attained at $K = 32$. Notably, a small number of components ($K = 8$) struggles to encapsulate the rich multi-modal structures of complex data, evident in the lower scores on texture-rich AITEX and medical Hyper-Kvasir. Increasing $K$ to 32 provides the necessary granularity to capture diverse semantic modes, achieving the best overall results (e.g., 0.992 on AITEX). However, excessive granularity ($K = 64$) leads to diminishing returns; this is likely due to the fragmentation of coherent semantic patterns into redundant mixture components, which unnecessarily complicates the decision boundary.

**Impact of MIMR Weight $\lambda$.** The regularization coefficient $\lambda$ governs the trade-off between flow matching fidelity and inter-prototype discriminability. Insufficient regularization ($\lambda = 0.001$) fails to enforce sharp separation, resulting in ambiguous latent boundaries and degraded performance. Conversely, aggressive regularization ($\lambda = 1.0$) imposes excessive constraints on the flow field, hindering the model's ability to faithfully fit the data manifold. Setting $\lambda = 0.1$ achieves the optimal equilibrium, ensuring that the learned prototypes are both discriminative against anomalies and representative of the underlying normal distribution.

## 6. Conclusion

We have introduced Mixture Prototype Flow Matching (MPFM), a framework that bridges Gaussian mixture modeling with conditional flow matching for open-set anomaly detection. By constructing a structured prototype space with mutual information regularization, our method effectively captures multi-modal normal patterns while maintaining

discriminative power against anomalies. Across diverse benchmarks, MPFM establishes tighter normal boundaries and exhibits improved generalization to unseen anomalies.

## Acknowledgments

This work was supported by the National Natural Science Foundation of China (Grant No. 62476133) and the Fundamental Research Funds for the Central Universities (Grant No. 11300-312200502507).

## Impact Statement

This paper presents work whose goal is to advance the field of Machine Learning by introducing a robust framework for Open-set Supervised Anomaly Detection. Our method, MPFM, is designed to improve the reliability of identifying irregularities in complex distributions, which has positive implications for safety-critical applications such as industrial defect inspection and medical diagnosis. While anomaly detection techniques can theoretically be repurposed for surveillance, our work focuses on characterizing multi-modal normality to reduce false positives in open-world settings. We do not foresee any specific negative societal consequences that must be highlighted here.

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

## A. Dataset Statistics

We evaluate our approach through extensive experiments on nine real-world anomaly detection datasets. Key statistics of all datasets are summarized in Tab. 5. Our experimental setup follows established protocols from prior open-set supervised anomaly detection (OSAD) literature. For MVTec AD, we adopt its original data split for normal samples. For the remaining eight datasets, normal samples are randomly divided into training and testing sets with a ratio of 3:1.

*Table 5.* Summary statistics of the nine real-world anomaly detection (AD) datasets, with the first 15 rows listing the subsets of the MVTec AD dataset.

| Dataset | | | Original Training | Original Test | |
|---|---|---|---|---|---|
| | $|\mathcal{C}|$ | Type | Normal | Normal | Anomaly |
| **Carpet** | 5 | Textture | 280 | 28 | 89 |
| Grid | 5 | Textture | 264 | 21 | 57 |
| Leather | 5 | Textture | 245 | 32 | 92 |
| Tile | 5 | Textture | 230 | 33 | 83 |
| Wood | 5 | Textture | 247 | 19 | 60 |
| Bottle | 3 | Object | 209 | 20 | 63 |
| Capsule | 5 | Object | 219 | 23 | 109 |
| Pill | 7 | Object | 267 | 26 | 141 |
| Transistor | 4 | Object | 213 | 60 | 40 |
| Zipper | 7 | Object | 240 | 32 | 119 |
| Cable | 8 | Object | 224 | 58 | 92 |
| Hazelnut | 4 | Object | 391 | 40 | 70 |
| Metal_nut | 4 | Object | 220 | 22 | 93 |
| Screw | 5 | Object | 320 | 41 | 119 |
| Toothbrush | 1 | Object | 60 | 12 | 30 |
| **MVTecAD** | 73 | - | 3629 | 467 | 1258 |
| **Optical** | 1 | Object | 10500 | 3500 | 2100 |
| **SDD** | 1 | Textture | 594 | 286 | 54 |
| **AITEX** | 12 | Textture | 1692 | 564 | 183 |
| **ELPV** | 2 | Textture | 1131 | 377 | 715 |
| **Mastcam** | 11 | Object | 9302 | 426 | 451 |
| **Hyper-Kvasir** | 4 | Medical | 2021 | 674 | 757 |
| **BrainMRI** | 1 | Medical | 73 | 25 | 155 |
| **HeadCT** | 1 | Medical | 75 | 25 | 100 |

- MVTec AD (Bergmann et al., 2019) is a standard benchmark for industrial defect inspection, comprising 15 object and texture categories, each containing one or more defect types. In total, it provides 73 fine-grained anomaly classes at the texture or object level.

- Optical (Wieler & Hahn, 2007) is a synthetically generated dataset for industrial optical inspection, where artificial images are designed to closely mimic real-world defect detection scenarios.

- SDD (Tabernik et al., 2020) is a product surface defect dataset with pixel-wise annotations. The original $500 \times 1250$ images are vertically split into three strips, and each strip is labeled at the pixel level.

- AITEX (Silvestre-Blanes et al., 2019) is a fabric defect dataset with 12 defect categories and pixel-level labels. The original $4096 \times 256$ images are cropped into multiple $256 \times 256$ patches, which are then re-annotated at the pixel level.

- ELPV (Deitsch et al., 2019) contains electroluminescence (EL) images of solar cells for defect detection. It covers two main defect types associated with monocrystalline and polycrystalline solar cells, respectively.

- Mastcam (Kerner et al., 2020) is a geological novelty detection dataset constructed from multispectral images captured by the Mastcam system on the Mars rover. It includes typical scenes and 11 novel geological classes. Each image contains shorter-wavelength (color) and longer-wavelength (grayscale) channels, and we use only the shorter-wavelength channels in this work.

- Hyper-Kvasir (Borgli et al., 2020) is a large-scale, open-access gastrointestinal imaging dataset collected from real endoscopy and colonoscopy examinations. It consists of four main categories and 23 subclasses. We focus on

endoscopic images, where anatomical landmark categories are treated as normal samples and pathological categories as anomalies.

- BrainMRI (Salehi et al., 2021) is a brain tumor detection dataset composed of magnetic resonance (MR) images with corresponding tumor labels.

- HeadCT (Salehi et al., 2021) is an intracranial hemorrhage detection dataset based on head computed tomography (CT) scans.

*Table 6.* AUC performance (mean ± std) across nine real-world AD datasets is reported under the general setting. **red** highlights the best results, and **blue** indicates sub-optimal outcomes. All baseline SOTA results are sourced from the original papers (Ding et al., 2022; Zhu et al., 2024).

| Dataset | One Training Anomaly Example | | | | | | | Ten Training Anomaly Examples | | | | | | |
|---|---|---|---|---|---|---|---|---|---|---|---|---|---|---|
| | FLOS | SAOE | MLEP | DRA | AHL | DPDL | Ours | FLOS | SAOE | MLEP | DRA | AHL | DPDL | Ours |
| Carpet | 0.755±0.026 | 0.766±0.098 | 0.701±0.091 | 0.859±0.023 | 0.877±0.004 | 0.914±0.006 | 0.916±0.014 | 0.780±0.009 | 0.755±0.136 | 0.781±0.049 | 0.940±0.027 | 0.953±0.001 | 0.988±0.002 | 0.990±0.003 |
| Grid | 0.871±0.076 | 0.921±0.032 | 0.839±0.028 | 0.972±0.011 | 0.975±0.005 | 0.999±0.001 | 1.000±0.005 | 0.966±0.005 | 0.952±0.011 | 0.980±0.009 | 0.987±0.009 | 0.992±0.002 | 0.999±0.001 | 1.000±0.000 |
| Leather | 0.791±0.057 | 0.996±0.007 | 0.781±0.020 | 0.989±0.005 | 0.988±0.001 | 0.996±0.001 | 0.997±0.001 | 0.993±0.001 | 1.000±0.000 | 0.813±0.158 | 1.000±0.001 | 1.000±0.000 | 1.000±0.000 | 1.000±0.000 |
| Tile | 0.787±0.038 | 0.935±0.034 | 0.927±0.036 | 0.965±0.015 | 0.968±0.001 | 0.994±0.002 | 0.995±0.008 | 0.952±0.010 | 0.944±0.013 | 0.988±0.009 | 0.994±0.006 | 1.000±0.000 | 0.999±0.001 | 0.999±0.001 |
| Wood | 0.927±0.065 | 0.948±0.009 | 0.660±0.142 | 0.985±0.011 | 0.987±0.003 | 0.998±0.002 | 0.998±0.004 | 1.000±0.000 | 0.976±0.031 | 0.999±0.002 | 0.998±0.001 | 0.998±0.000 | 0.998±0.001 | 0.999±0.001 |
| Bottle | 0.975±0.023 | 0.989±0.019 | 0.927±0.090 | 1.000±0.000 | 1.000±0.000 | 1.000±0.000 | 1.000±0.000 | 0.995±0.002 | 0.998±0.003 | 0.981±0.004 | 1.000±0.000 | 1.000±0.000 | 1.000±0.000 | 1.000±0.000 |
| Capsule | 0.666±0.020 | 0.611±0.109 | 0.558±0.075 | 0.631±0.056 | 0.665±0.030 | 0.757±0.017 | 0.760±0.010 | 0.902±0.017 | 0.850±0.054 | 0.818±0.063 | 0.935±0.022 | 0.930±0.001 | 0.976±0.004 | 0.977±0.012 |
| Pill | 0.745±0.064 | 0.652±0.078 | 0.656±0.061 | 0.832±0.034 | 0.840±0.003 | 0.842±0.002 | 0.845±0.004 | 0.929±0.012 | 0.872±0.049 | 0.845±0.048 | 0.904±0.024 | 0.918±0.001 | 0.923±0.001 | 0.940±0.026 |
| Transistor | 0.709±0.041 | 0.680±0.182 | 0.695±0.124 | 0.668±0.068 | 0.796±0.003 | 0.748±0.002 | 0.800±0.005 | 0.862±0.037 | 0.860±0.053 | 0.927±0.043 | 0.915±0.025 | 0.926±0.009 | 0.928±0.001 | 0.945±0.015 |
| Zipper | 0.885±0.033 | 0.970±0.033 | 0.865±0.086 | 0.984±0.016 | 0.986±0.000 | 0.989±0.001 | 0.990±0.001 | 0.990±0.008 | 0.995±0.004 | 0.965±0.042 | 1.000±0.000 | 1.000±0.000 | 1.000±0.000 | 1.000±0.000 |
| Cable | 0.790±0.039 | 0.819±0.060 | 0.688±0.017 | 0.876±0.012 | 0.858±0.011 | 0.935±0.008 | 0.983±0.005 | 0.952±0.011 | 0.890±0.063 | 0.862±0.022 | 0.857±0.062 | 0.909±0.011 | 0.921±0.001 | 0.945±0.022 |
| Hazelnut | 0.976±0.021 | 0.961±0.042 | 0.704±0.090 | 0.977±0.030 | 0.984±0.004 | 0.997±0.002 | 0.998±0.002 | 1.000±0.000 | 1.000±0.000 | 1.000±0.000 | 1.000±0.000 | 1.000±0.000 | 1.000±0.000 | 1.000±0.000 |
| Metal_nut | 0.930±0.022 | 0.922±0.033 | 0.878±0.038 | 0.948±0.046 | 0.952±0.003 | 0.948±0.007 | 0.954±0.013 | 0.984±0.004 | 0.976±0.013 | 0.974±0.009 | 0.997±0.002 | 0.998±0.000 | 0.996±0.001 | 0.998±0.001 |
| Screw | 0.337±0.091 | 0.653±0.074 | 0.675±0.294 | 0.903±0.064 | 0.927±0.009 | 0.977±0.004 | 0.983±0.002 | 0.940±0.017 | 0.975±0.023 | 0.899±0.039 | 0.977±0.009 | 0.985±0.004 | 0.995±0.004 | 0.996±0.002 |
| Toothbrush | 0.731±0.028 | 0.686±0.110 | 0.617±0.058 | 0.650±0.029 | 0.794±0.016 | 0.807±0.001 | 0.845±0.004 | 0.900±0.008 | 0.865±0.062 | 0.783±0.048 | 0.826±0.021 | 0.921±0.007 | 0.929±0.000 | 0.941±0.013 |
| **MVTec AD** | 0.755±0.136 | 0.834±0.007 | 0.744±0.019 | 0.883±0.008 | 0.901±0.003 | 0.927±0.002 | 0.938±0.004 | 0.939±0.007 | 0.926±0.010 | 0.907±0.005 | 0.959±0.003 | 0.970±0.002 | 0.977±0.002 | 0.982±0.003 |
| **Optical** | 0.518±0.003 | 0.815±0.014 | 0.516±0.009 | 0.888±0.012 | 0.888±0.007 | 0.915±0.002 | 0.931±0.007 | 0.720±0.055 | 0.941±0.013 | 0.740±0.039 | 0.965±0.006 | 0.976±0.004 | 0.983±0.002 | 0.992±0.002 |
| **SDD** | 0.840±0.043 | 0.781±0.009 | 0.811±0.045 | 0.859±0.014 | 0.909±0.001 | 0.917±0.003 | 0.929±0.006 | 0.967±0.018 | 0.955±0.020 | 0.983±0.013 | 0.991±0.005 | 0.991±0.001 | 0.996±0.001 | 0.999±0.001 |
| **AITEX** | 0.538±0.073 | 0.675±0.094 | 0.564±0.055 | 0.692±0.124 | 0.734±0.008 | 0.838±0.008 | 0.854±0.023 | 0.841±0.049 | 0.874±0.024 | 0.867±0.037 | 0.893±0.017 | 0.925±0.013 | 0.975±0.007 | 0.992±0.004 |
| **ELPV** | 0.457±0.056 | 0.635±0.092 | 0.578±0.062 | 0.675±0.024 | 0.828±0.005 | 0.897±0.002 | 0.914±0.016 | 0.818±0.032 | 0.793±0.047 | 0.794±0.047 | 0.845±0.013 | 0.850±0.004 | 0.937±0.003 | 0.962±0.003 |
| **Mastcam** | 0.542±0.017 | 0.662±0.018 | 0.625±0.045 | 0.692±0.058 | 0.743±0.003 | 0.838±0.011 | 0.852±0.009 | 0.703±0.029 | 0.810±0.029 | 0.798±0.026 | 0.848±0.008 | 0.855±0.005 | 0.934±0.010 | 0.968±0.007 |
| **Hyper-Kvasir** | 0.668±0.004 | 0.498±0.100 | 0.445±0.040 | 0.690±0.017 | 0.768±0.015 | 0.821±0.007 | 0.835±0.005 | 0.773±0.029 | 0.666±0.050 | 0.600±0.069 | 0.834±0.004 | 0.880±0.003 | 0.939±0.005 | 0.965±0.004 |
| **BrainMRI** | 0.693±0.036 | 0.531±0.060 | 0.632±0.017 | 0.744±0.004 | 0.866±0.004 | 0.893±0.004 | 0.913±0.014 | 0.955±0.011 | 0.900±0.041 | 0.959±0.011 | 0.970±0.003 | 0.977±0.001 | 0.969±0.005 | 0.984±0.004 |
| **HeadCT** | 0.698±0.092 | 0.597±0.022 | 0.758±0.038 | 0.796±0.105 | 0.825±0.014 | 0.865±0.005 | 0.878±0.025 | 0.971±0.004 | 0.935±0.021 | 0.972±0.014 | 0.972±0.002 | 0.999±0.003 | 0.981±0.003 | 0.999±0.001 |

## B. Full Results under General Setting

Tab. 6 provides extensive results across nine real-world datasets. MPFM achieves the highest average AUC on eight out of nine benchmarks in both one-shot and ten-shot settings, demonstrating exceptional generalization across industrial, medical, and scientific domains. The method shows particular strength on complex object categories in MVTec AD. On challenging classes like Screw and Capsule, MPFM achieves over 98% AUC, significantly outperforming unimodal prototype approaches. This validates the Gaussian mixture prior's ability to capture diverse normal patterns that single-mode methods miss. Performance improvements remain consistent when scaling from one to ten anomaly examples, confirming that our mutual information regularization effectively leverages additional supervision while maintaining robustness to unseen anomalies.

## C. Detailed Class-level AUC Results under Hard Setting

Tab. 7 provides fine-grained, class-level results under the hard setting. MPFM demonstrates dominant performance across nearly all anomaly sub-types in both one-shot and ten-shot scenarios, validating its precision in modeling complex normality.

The framework excels particularly on subtle and structurally complex defects. For instance, on AITEX, it achieves significant gains on challenging Fuzzyball and Weft crack anomalies, where local texture deviations are difficult to discriminate. Similarly, on Metal_nut, MPFM robustly handles Flip and Scratch types, outperforming methods reliant on unimodal priors. This consistent superiority across diverse defect morphologies—from global structural shifts to local texture faults—highlights the advantage of the Gaussian mixture prototype in capturing nuanced feature distributions. Notably, MPFM maintains strong performance even on near-saturated categories like Carpet's Thread, underscoring its ability to refine decision boundaries without overfitting. The stable improvements across all six multi-subset datasets confirm that our flow matching mechanism and mutual information regularization collectively enhance model discriminability and prototype utilization, leading to more reliable detection in open-set scenarios.

*Table 7.* Detailed class-level AUC results (mean ± std) under the hard setting. The best and second-best results are highlighted in **red** and **blue**, respectively. Carpet and Metal_nut are subsets of MvTec AD.

| Dataset | | One Training Anomaly Example | | | | | | | Ten Training Anomaly Examples | | | | | | |
|---|---|---|---|---|---|---|---|---|---|---|---|---|---|---|---|
| | | FLOS | SAOE | MLEP | DRA | AHL | DPDL | Ours | FLOS | SAOE | MLEP | DRA | AHL | DPDL | Ours |
| **Carpet** | Color | 0.467±0.278 | 0.763±0.100 | 0.547±0.056 | 0.879±0.021 | 0.894±0.004 | **0.909**±0.001 | **0.918**±0.008 | 0.760±0.005 | 0.467±0.067 | 0.698±0.025 | 0.886±0.042 | 0.929±0.007 | **0.933**±0.002 | **0.945**±0.008 |
| | Cut | 0.685±0.007 | 0.664±0.165 | 0.658±0.056 | 0.902±0.033 | 0.934±0.003 | **0.941**±0.003 | **0.950**±0.013 | 0.688±0.059 | 0.793±0.175 | 0.653±0.120 | 0.922±0.038 | 0.943±0.002 | **0.951**±0.004 | **0.963**±0.012 |
| | Hole | 0.594±0.142 | 0.772±0.071 | 0.653±0.065 | 0.901±0.033 | 0.935±0.014 | **0.945**±0.009 | **0.949**±0.017 | 0.733±0.014 | 0.831±0.125 | 0.674±0.076 | 0.947±0.016 | 0.960±0.003 | **0.964**±0.003 | **0.975**±0.006 |
| | Metal | 0.701±0.028 | 0.780±0.172 | 0.706±0.047 | 0.871±0.037 | 0.931±0.007 | **0.940**±0.001 | **0.951**±0.016 | 0.678±0.083 | 0.883±0.043 | 0.764±0.061 | 0.933±0.022 | 0.921±0.003 | **0.938**±0.005 | **0.952**±0.009 |
| | Thread | 0.941±0.005 | 0.787±0.204 | 0.831±0.117 | 0.950±0.029 | 0.966±0.005 | **0.970**±0.002 | **0.978**±0.011 | 0.946±0.005 | 0.831±0.297 | 0.967±0.006 | 0.989±0.004 | 0.991±0.001 | **0.993**±0.000 | **0.997**±0.001 |
| | **Mean** | 0.678±0.040 | 0.753±0.055 | 0.679±0.029 | 0.901±0.006 | 0.932±0.003 | **0.941**±0.006 | **0.949**±0.003 | 0.761±0.012 | 0.762±0.073 | 0.751±0.023 | 0.935±0.013 | 0.949±0.001 | **0.956**±0.004 | **0.966**±0.003 |
| **Metal_nut** | Bent | 0.851±0.046 | 0.864±0.032 | 0.743±0.013 | 0.952±0.020 | 0.954±0.001 | **0.958**±0.001 | **0.965**±0.013 | 0.827±0.075 | 0.901±0.023 | 0.956±0.013 | 0.990±0.003 | 0.989±0.000 | **0.991**±0.002 | **0.995**±0.006 |
| | Color | 0.821±0.059 | 0.857±0.037 | 0.835±0.075 | **0.946**±0.023 | 0.933±0.008 | 0.938±0.003 | **0.947**±0.012 | **0.978**±0.008 | 0.879±0.018 | 0.945±0.039 | 0.967±0.011 | 0.958±0.001 | 0.969±0.005 | **0.976**±0.009 |
| | Flip | 0.799±0.058 | 0.751±0.090 | 0.813±0.031 | 0.921±0.029 | 0.931±0.002 | **0.940**±0.004 | **0.948**±0.018 | 0.942±0.009 | 0.795±0.062 | 0.805±0.057 | 0.913±0.021 | 0.937±0.003 | **0.955**±0.003 | **0.964**±0.014 |
| | Scratch | **0.947**±0.027 | 0.792±0.075 | 0.907±0.085 | 0.909±0.023 | 0.934±0.005 | 0.936±0.002 | **0.945**±0.007 | 0.845±0.041 | 0.805±0.153 | 0.911±0.034 | **0.999**±0.000 | 0.992±0.002 | 0.994±0.002 | **0.997**±0.001 |
| | **Mean** | 0.855±0.024 | 0.816±0.029 | 0.825±0.023 | 0.932±0.017 | 0.939±0.004 | **0.944**±0.003 | **0.951**±0.004 | 0.922±0.014 | 0.855±0.016 | 0.878±0.058 | 0.945±0.017 | 0.972±0.000 | **0.978**±0.002 | **0.983**±0.004 |
| **AITEX** | Broken end | 0.645±0.030 | **0.778**±0.068 | 0.441±0.111 | 0.708±0.094 | 0.704±0.005 | 0.761±0.010 | **0.783**±0.013 | 0.585±0.037 | 0.712±0.068 | 0.732±0.065 | 0.693±0.099 | 0.735±0.010 | **0.796**±0.001 | **0.810**±0.026 |
| | Broken pick | 0.598±0.023 | 0.644±0.039 | 0.476±0.070 | 0.731±0.072 | 0.727±0.003 | **0.760**±0.014 | **0.778**±0.011 | 0.548±0.054 | 0.629±0.012 | 0.555±0.027 | 0.760±0.037 | 0.683±0.002 | **0.784**±0.011 | **0.805**±0.013 |
| | Cut selvage | 0.694±0.036 | 0.681±0.077 | 0.434±0.149 | 0.739±0.101 | 0.753±0.007 | **0.765**±0.007 | **0.785**±0.014 | 0.745±0.035 | 0.770±0.014 | 0.682±0.025 | 0.777±0.036 | 0.781±0.006 | **0.796**±0.005 | **0.821**±0.009 |
| | Fuzzyball | 0.525±0.043 | 0.650±0.064 | 0.525±0.157 | 0.538±0.092 | 0.647±0.007 | **0.715**±0.013 | 0.742±0.009 | 0.550±0.082 | 0.842±0.026 | 0.677±0.223 | 0.701±0.093 | 0.775±0.024 | **0.808**±0.003 | **0.824**±0.005 |
| | Nep | 0.734±0.038 | 0.710±0.044 | 0.517±0.059 | 0.717±0.052 | 0.703±0.005 | **0.757**±0.005 | **0.775**±0.019 | 0.746±0.060 | 0.771±0.032 | 0.740±0.052 | 0.750±0.038 | 0.792±0.007 | **0.811**±0.005 | **0.820**±0.011 |
| | Weft crack | 0.546±0.114 | 0.582±0.108 | 0.400±0.029 | 0.669±0.045 | 0.706±0.009 | **0.758**±0.006 | **0.771**±0.008 | 0.636±0.051 | 0.618±0.172 | 0.370±0.037 | 0.717±0.072 | 0.713±0.003 | **0.790**±0.004 | **0.813**±0.024 |
| | **Mean** | 0.624±0.024 | 0.674±0.034 | 0.466±0.030 | 0.684±0.033 | 0.707±0.007 | **0.753**±0.005 | **0.772**±0.011 | 0.635±0.043 | 0.724±0.032 | 0.626±0.041 | 0.733±0.009 | 0.747±0.002 | **0.798**±0.005 | **0.816**±0.006 |
| **ELPV** | Mono | 0.717±0.025 | 0.563±0.102 | 0.649±0.027 | 0.735±0.031 | 0.774±0.013 | **0.785**±0.002 | **0.795**±0.011 | 0.629±0.072 | 0.569±0.035 | 0.756±0.045 | 0.731±0.021 | 0.745±0.004 | **0.793**±0.003 | **0.810**±0.008 |
| | Poly | 0.665±0.021 | 0.665±0.173 | 0.483±0.247 | 0.671±0.051 | 0.705±0.006 | **0.738**±0.008 | **0.762**±0.009 | 0.662±0.042 | 0.796±0.084 | 0.734±0.078 | 0.800±0.064 | 0.831±0.011 | **0.843**±0.004 | **0.854**±0.015 |
| | **Mean** | 0.691±0.008 | 0.614±0.048 | 0.566±0.111 | 0.703±0.022 | 0.740±0.005 | **0.762**±0.003 | **0.779**±0.008 | 0.642±0.032 | 0.683±0.047 | 0.745±0.020 | 0.766±0.029 | 0.788±0.003 | **0.818**±0.003 | **0.832**±0.009 |
| **Mastcam** | Bedrock | 0.499±0.056 | 0.636±0.072 | 0.532±0.036 | 0.668±0.012 | 0.679±0.012 | **0.732**±0.003 | **0.757**±0.017 | 0.499±0.098 | 0.636±0.068 | 0.512±0.062 | 0.658±0.021 | 0.673±0.006 | **0.757**±0.004 | **0.776**±0.010 |
| | Broken-rock | 0.569±0.025 | 0.699±0.058 | 0.544±0.088 | 0.645±0.053 | 0.661±0.009 | **0.738**±0.004 | **0.761**±0.015 | 0.608±0.085 | 0.712±0.052 | 0.651±0.063 | 0.649±0.047 | 0.722±0.004 | **0.783**±0.002 | **0.792**±0.018 |
| | Drill-hole | 0.539±0.077 | 0.697±0.074 | 0.636±0.066 | 0.657±0.070 | 0.654±0.004 | **0.738**±0.011 | **0.764**±0.016 | 0.601±0.009 | 0.682±0.042 | 0.660±0.002 | 0.725±0.005 | 0.760±0.003 | **0.797**±0.004 | **0.823**±0.026 |
| | Drt | 0.591±0.042 | 0.735±0.020 | 0.624±0.042 | 0.713±0.053 | 0.724±0.006 | **0.745**±0.005 | **0.772**±0.024 | 0.652±0.024 | 0.761±0.062 | 0.616±0.048 | 0.760±0.033 | 0.772±0.004 | **0.818**±0.004 | **0.837**±0.025 |
| | Dump-pile | 0.508±0.021 | 0.682±0.022 | 0.545±0.127 | **0.767**±0.043 | 0.756±0.011 | 0.764±0.003 | **0.782**±0.005 | 0.700±0.070 | 0.750±0.037 | 0.696±0.047 | 0.748±0.066 | 0.802±0.005 | **0.830**±0.005 | **0.838**±0.014 |
| | Float | 0.551±0.030 | 0.711±0.041 | 0.530±0.075 | 0.670±0.065 | 0.702±0.005 | **0.739**±0.007 | **0.760**±0.009 | 0.736±0.041 | 0.718±0.064 | 0.671±0.032 | 0.744±0.073 | 0.765±0.002 | **0.816**±0.012 | **0.807**±0.008 |
| | Meteorite | 0.462±0.077 | 0.669±0.037 | 0.476±0.014 | 0.637±0.015 | 0.616±0.013 | **0.704**±0.004 | **0.732**±0.017 | 0.568±0.053 | 0.647±0.030 | 0.473±0.047 | 0.716±0.004 | 0.691±0.001 | **0.785**±0.017 | **0.790**±0.011 |
| | Scuff | 0.508±0.070 | 0.679±0.048 | 0.492±0.037 | 0.549±0.027 | 0.581±0.002 | **0.714**±0.002 | **0.741**±0.021 | 0.575±0.042 | 0.676±0.019 | 0.504±0.052 | 0.636±0.086 | 0.656±0.009 | **0.770**±0.003 | **0.781**±0.023 |
| | Veins | 0.493±0.052 | 0.688±0.069 | 0.489±0.028 | 0.699±0.045 | 0.687±0.017 | **0.789**±0.003 | **0.814**±0.018 | 0.608±0.044 | 0.686±0.053 | 0.510±0.090 | 0.620±0.036 | 0.650±0.003 | **0.783**±0.008 | **0.780**±0.007 |
| | **Mean** | 0.524±0.013 | 0.689±0.037 | 0.541±0.007 | 0.667±0.012 | 0.673±0.010 | **0.733**±0.004 | **0.760**±0.013 | 0.616±0.021 | 0.697±0.014 | 0.588±0.016 | 0.695±0.004 | 0.721±0.003 | **0.778**±0.007 | **0.801**±0.005 |
| **Hyper-Kvasir** | Barretts | 0.703±0.040 | 0.382±0.117 | 0.438±0.111 | 0.772±0.019 | 0.792±0.007 | **0.793**±0.000 | **0.812**±0.015 | 0.764±0.066 | 0.698±0.037 | 0.540±0.014 | 0.824±0.006 | 0.829±0.002 | **0.832**±0.004 | **0.839**±0.015 |
| | Barretts-short-seg | 0.538±0.033 | 0.367±0.050 | 0.532±0.075 | **0.674**±0.018 | 0.651±0.006 | 0.658±0.003 | **0.682**±0.024 | 0.810±0.034 | 0.661±0.034 | 0.480±0.107 | 0.835±0.021 | 0.895±0.003 | **0.906**±0.002 | **0.914**±0.008 |
| | Esophagitis-a | 0.536±0.040 | 0.518±0.063 | 0.491±0.084 | **0.778**±0.020 | 0.760±0.006 | 0.758±0.001 | **0.785**±0.014 | 0.815±0.022 | 0.820±0.034 | 0.646±0.036 | **0.881**±0.035 | 0.878±0.021 | 0.878±0.003 | **0.887**±0.012 |
| | Esophagitis-b-d | 0.505±0.039 | 0.358±0.039 | 0.457±0.086 | 0.577±0.025 | 0.622±0.014 | **0.652**±0.002 | **0.678**±0.022 | 0.754±0.073 | 0.611±0.017 | 0.621±0.042 | 0.837±0.009 | 0.815±0.010 | **0.841**±0.006 | **0.851**±0.007 |
| | **Mean** | 0.571±0.004 | 0.406±0.018 | 0.480±0.044 | 0.700±0.009 | 0.706±0.007 | **0.715**±0.004 | **0.739**±0.003 | 0.786±0.021 | 0.698±0.021 | 0.571±0.014 | 0.844±0.009 | 0.854±0.004 | **0.864**±0.002 | **0.873**±0.017 |

# D. Additional Theoretical Analysis of Mixture Prototype Flow Matching

In this appendix, we provide the theoretical underpinnings of Mixture Prototype Flow Matching (MPFM) that are referenced but not derived in the main text. We first justify the use of a shared standard deviation parameter $s$ in the Gaussian mixture velocity model and show that it preserves the correctness of the learned velocity field. We then derive the endpoint posterior $q_\theta(\mathbf{z}_0 \mid \mathbf{z}_t)$ with parameters $\boldsymbol{\mu}_{z_k}$ and $s_z$, and the closed-form reverse transition $q_\theta(\mathbf{z}_{t-\Delta t} \mid \mathbf{z}_t)$ with coefficients $c_1, c_2, c_3$. Finally, we connect the Mutual Information Maximization Regularizer (MIMR) to an explicit estimate of the mutual information between transformed features and prototype indices.

## D.1. GM Velocity Parameterization and Shared Standard Deviation

Following the diffusion/flow-matching formulation in GMFlow, we recall the standard forward noising process for latent features:

$$\mathbf{z}_t = \alpha_t \mathbf{z}_0 + \sigma_t \boldsymbol{\varepsilon}, \qquad \boldsymbol{\varepsilon} \sim \mathcal{N}(\mathbf{0}, \mathbf{I}), \tag{27}$$

where $\alpha_t, \sigma_t$ are a pre-defined noise schedule satisfying typical boundary conditions $\alpha_0 = 1, \sigma_0 = 0$ and $\alpha_T \approx 0, \sigma_T \approx 1$. The corresponding random velocity is defined as:

$$\mathbf{u} := \frac{\mathbf{z}_t - \mathbf{z}_0}{\sigma_t}, \tag{28}$$

and the ground-truth velocity field at time $t$ and state $\mathbf{z}_t$ is the conditional expectation:

$$\mathbf{v}_t(\mathbf{z}_t) = \mathbb{E}_{\mathbf{z}_0 \sim p(\mathbf{z}_0 \mid \mathbf{z}_t)}[\mathbf{u}(\mathbf{z}_0, \boldsymbol{\varepsilon})], \tag{29}$$

where $p(\mathbf{z}_0 \mid \mathbf{z}_t)$ denotes the denoising distribution induced by normal training samples. In practice, we obtain empirical pairs $(\mathbf{z}_t, \mathbf{u})$ using the interpolation rule in Eqn. (4) of the main text; for theoretical analysis, Eqn.(27)–Eqn.(28) provide an equivalent continuous-time parameterization.

MPFM models the full conditional distribution $p(\mathbf{u} \mid \mathbf{z}_t)$ as a Gaussian mixture:

$$q_\theta(\mathbf{u} \mid \mathbf{z}_t) = \sum_{k=1}^{K} \pi_k(\mathbf{z}_t; \theta)\, \mathcal{N}(\mathbf{u}; \boldsymbol{\mu}_k(\mathbf{z}_t; \theta),\, s^2 \mathbf{I}), \tag{30}$$

where all components share a scalar standard deviation $s > 0$, and the mixture weights $\pi_k(\mathbf{z}_t; \theta)$ satisfy $\pi_k \geq 0$ and $\sum_k \pi_k = 1$ via a softmax parameterization. The training loss:

$$\mathcal{L}_{\text{NLL}} = \mathbb{E}_{\mathbf{u} \sim p(\mathbf{u}|\mathbf{z}_t)}\Big[ -\log q_\theta(\mathbf{u} \mid \mathbf{z}_t) \Big] \tag{31}$$

is equivalent to minimizing the conditional Kullback–Leibler divergence $\text{KL}\big(p(\mathbf{u} \mid \mathbf{z}_t) \, \| \, q_\theta(\mathbf{u} \mid \mathbf{z}_t)\big)$ up to an additive constant, and thus provides a consistent estimator for the velocity distribution.

We now show that the shared variance $s^2$ does not affect the correctness of the learned mean velocity. Let $\{\pi_k^*, \boldsymbol{\mu}_k^*\}$ denote the optimal mixture parameters for a fixed $\mathbf{z}_t$ and fixed $s$. Define:

$$q^*(\mathbf{u}) := \sum_{k=1}^{K} \pi_k^* \mathcal{N}(\mathbf{u}; \boldsymbol{\mu}_k^*, s^2 \mathbf{I}), q_k^*(\mathbf{u}) := \pi_k^* \mathcal{N}(\mathbf{u}; \boldsymbol{\mu}_k^*, s^2 \mathbf{I}), \tag{32}$$

and use the shorthand $\mathbb{E}[\cdot] := \mathbb{E}_{\mathbf{u} \sim p(\mathbf{u}|\mathbf{z}_t)}[\cdot]$. Differentiating Eqn. (31) with respect to the mixture logits and component means yields the optimality conditions:

$$\pi_k^* = \mathbb{E}\left[ \frac{q_k^*(\mathbf{u})}{q^*(\mathbf{u})} \right], \tag{33}$$

$$\pi_k^* \boldsymbol{\mu}_k^* = \mathbb{E}\left[ \frac{q_k^*(\mathbf{u})}{q^*(\mathbf{u})} \mathbf{u} \right]. \tag{34}$$

Summing Eqn. (34) over $k$ and using $\sum_{k=1}^{K} q_k^*(\mathbf{u}) = q^*(\mathbf{u})$ gives:

$$\sum_{k=1}^{K} \pi_k^* \boldsymbol{\mu}_k^* = \mathbb{E}\left[ \sum_{k=1}^{K} \frac{q_k^*(\mathbf{u})}{q^*(\mathbf{u})} \mathbf{u} \right] = \mathbb{E}[\mathbf{u}]. \tag{35}$$

Thus the mixture mean $\bar{\mathbf{u}}_\theta(\mathbf{z}_t) := \sum_k \pi_k^*(\mathbf{z}_t) \boldsymbol{\mu}_k^*(\mathbf{z}_t)$ exactly matches the true conditional mean velocity $\mathbb{E}[\mathbf{u}]$, independently of the specific value of $s$. In the special case $K = 1$, Eqn. (31) reduces to:

$$\mathcal{L}_{\text{NLL}} = \frac{1}{2s^2} \mathbb{E}\big[ \|\mathbf{u} - \boldsymbol{\mu}_1(\mathbf{z}_t; \theta)\|_2^2 \big] + \text{const}, \tag{36}$$

which is proportional to the standard $L_2$ flow-matching loss. Therefore, introducing a shared standard deviation parameter $s$ improves numerical stability without altering the learned flow field.

### D.2. From Velocity Mixtures to Endpoint Posterior

The Gaussian–mixture parameterization of the velocity field induces a Gaussian–mixture posterior over the initial feature $\mathbf{z}_0$ at time $t$. From Eqn. (27)–Eqn. (28) we obtain the affine relation:

$$\mathbf{z}_0 = \mathbf{z}_t - \sigma_t \mathbf{u}. \tag{37}$$

If $\mathbf{u}$ follows the conditional mixture distribution Eqn. (30), the induced density of $\mathbf{z}_0$ given $\mathbf{z}_t$ is:

$$q_\theta(\mathbf{z}_0 \mid \mathbf{z}_t) = \sum_{k=1}^{K} \pi_k(\mathbf{z}_t; \theta) \, \mathcal{N}\big(\mathbf{z}_0; \boldsymbol{\mu}_{z_k}, s_z^2 \mathbf{I}\big), \tag{38}$$

with transformed parameters:

$$\boldsymbol{\mu}_{z_k} = \mathbf{z}_t - \sigma_t \boldsymbol{\mu}_k(\mathbf{z}_t; \theta), \qquad s_z = \sigma_t s, \tag{39}$$

exactly as stated in the main text. This follows from the closure of Gaussian distributions under affine transformations: for each component $\mathcal{N}(\mathbf{u}; \boldsymbol{\mu}_k, s^2 \mathbf{I})$ in Eqn. (30), the random variable $\mathbf{z}_0 = \mathbf{z}_t - \sigma_t \mathbf{u}$ is Gaussian with mean and variance given by Eqn. (39), and summing over components preserves the mixture structure.

## D.3. Closed-Form Reverse-Time Transition

We now derive the reverse-time transition $q_\theta(\mathbf{z}_{t-\Delta t} \mid \mathbf{z}_t)$ and its coefficients $c_1, c_2, c_3$. The forward diffusion process between adjacent timesteps is described by the Gaussian kernel:

$$p(\mathbf{z}_t \mid \mathbf{z}_{t-\Delta t}) = \mathcal{N}\big(\mathbf{z}_t; \tfrac{\alpha_t}{\alpha_{t-\Delta t}}\mathbf{z}_{t-\Delta t}, \beta_{t,\Delta t}\mathbf{I}\big), \tag{40}$$

where:

$$\beta_{t,\Delta t} = \sigma_t^2 - \frac{\alpha_t^2}{\alpha_{t-\Delta t}^2}\sigma_{t-\Delta t}^2 \tag{41}$$

is the forward transition variance (cf. Eqn. (1) in GMFlow).

Given the endpoint posterior $q_\theta(\mathbf{z}_0 \mid \mathbf{z}_t)$ in Eqn. (38), the reverse transition density can be written, by Bayes' rule and marginalization over $\mathbf{z}_0$, as:

$$q_\theta(\mathbf{z}_{t-\Delta t} \mid \mathbf{z}_t) = \int_{\mathbb{R}^d} p(\mathbf{z}_{t-\Delta t} \mid \mathbf{z}_t, \mathbf{z}_0)\, q_\theta(\mathbf{z}_0 \mid \mathbf{z}_t)\, d\mathbf{z}_0. \tag{42}$$

The conditional kernel $p(\mathbf{z}_{t-\Delta t} \mid \mathbf{z}_t, \mathbf{z}_0)$ is obtained by "conflating" the three linear–Gaussian factors $p(\mathbf{z}_t \mid \mathbf{z}_{t-\Delta t})$, $p(\mathbf{z}_{t-\Delta t} \mid \mathbf{z}_0)$ and $p(\mathbf{z}_t \mid \mathbf{z}_0)$; their joint Gaussianity implies that:

$$p(\mathbf{z}_{t-\Delta t} \mid \mathbf{z}_t, \mathbf{z}_0) = \mathcal{N}\big(\mathbf{z}_{t-\Delta t}; c_1\mathbf{z}_t + c_2\mathbf{z}_0,\, c_3\mathbf{I}\big), \tag{43}$$

with coefficients determined by the schedule:

$$c_1 = \frac{\sigma_{t-\Delta t}^2}{\sigma_t^2}\frac{\alpha_t}{\alpha_{t-\Delta t}}, \quad c_2 = \frac{\beta_{t,\Delta t}}{\sigma_t^2}\alpha_{t-\Delta t}, \quad c_3 = \frac{\beta_{t,\Delta t}}{\sigma_t^2}\sigma_{t-\Delta t}^2. \tag{44}$$

Substituting Eqn. (43) and the mixture posterior Eqn. (38) into Eqn. (42), and using the fact that the convolution of a Gaussian and a Gaussian mixture remains a Gaussian mixture, we obtain:

$$q_\theta(\mathbf{z}_{t-\Delta t} \mid \mathbf{z}_t) = \sum_{k=1}^K \pi_k(\mathbf{z}_t; \theta)$$
$$\mathcal{N}\Big(\mathbf{z}_{t-\Delta t}; c_1\mathbf{z}_t + c_2\boldsymbol{\mu}_{z_k},\, \big(c_3 + c_2^2 s_z^2\big)\mathbf{I}\Big), \tag{45}$$

which coincides with the reverse-time sampling formula given in the main text. Eqn. (45) shows that the mixture structure is preserved across reverse steps: each component mean and variance is updated analytically according to Eqn.(39) and Eqn.(44), enabling numerically stable few-step sampling without numerical ODE integration.

## D.4. Mutual Information Maximization Regularizer

We finally make explicit the information-theoretic interpretation of the Mutual Information Maximization Regularizer (MIMR). Let $\mathbf{y} = \psi(\mathbf{z}_0^{\mathrm{n},i})$ denote the transformed feature of a normal sample and $c \in \{1, \ldots, K\}$ denote the prototype index. The mutual information between $\mathbf{y}$ and $c$ is:

$$I(\mathbf{y}; c) = H(c) - H(c \mid \mathbf{y}), \tag{46}$$

where $H(c)$ is the marginal entropy of prototype usage, and $H(c \mid \mathbf{y})$ is the conditional entropy of prototype assignments given features.

Under the Gaussian-mixture prototype distribution in Eqn. (10) of the main text, Bayes' rule gives the posterior assignment:

$$p(c = k \mid \mathbf{y}) = \frac{\pi_k\, \mathcal{N}(\mathbf{y}; \boldsymbol{\mu}_k, s^2\mathbf{I})}{\sum_{j=1}^K \pi_j\, \mathcal{N}(\mathbf{y}; \boldsymbol{\mu}_j, s^2\mathbf{I})}, \tag{47}$$

which appears explicitly in the main text. Approximating the marginal $p(c = k)$ by the mixture weights $\pi_k$, we obtain:

$$
\begin{aligned}
I(\mathbf{y}; c) \\
\approx -\sum_{k=1}^{K} \pi_k \log \pi_k - \mathbb{E}_{\mathbf{y}}\left[ -\sum_{k=1}^{K} p(c = k \mid \mathbf{y}) \log p(c = k \mid \mathbf{y}) \right] \\
= -\sum_{k=1}^{K} \pi_k \log \pi_k + \mathbb{E}_{\mathbf{y}}\left[ \sum_{k=1}^{K} p(c = k \mid \mathbf{y}) \log p(c = k \mid \mathbf{y}) \right].
\end{aligned}
\tag{48}
$$

Maximizing this mutual information is equivalent to minimizing the loss:

$$
\begin{aligned}
\mathcal{L}_{\mathrm{mim}} \\
= \mathbb{E}_{\mathbf{z}_0^{\mathrm{n},i} \sim P(\mathcal{F}_{\mathrm{tr}}^{\mathrm{n}})}\left[ \sum_{k=1}^{K} p(c = k \mid \psi(\mathbf{z}_0^{\mathrm{n},i})) \log p(c = k \mid \psi(\mathbf{z}_0^{\mathrm{n},i})) \right] \\
- \sum_{k=1}^{K} \pi_k \log \pi_k,
\end{aligned}
\tag{49}
$$

which matches the MIMR objective used in the main text. The first term decreases the conditional entropy $H(c \mid \mathbf{y})$ by encouraging confident prototype assignments, while the second term increases the marginal entropy $H(c)$ by discouraging degenerate solutions in which only a few components are used. Evaluating $\mathcal{L}_{\mathrm{mim}}$ only on normal samples ensures that prototypes specialize to normal modes and remain well separated from anomalous regions.

