# OpenReview forum: "Mixture Prototype Flow Matching for Open-Set Supervised Anomaly Detection"
_ICML.cc/2026/Conference — ICML 2026 regular_

### Official Review · Reviewer_7JzR · 2026-03-10

**Soundness:** 3
**Presentation:** 3
**Significance:** 3
**Originality:** 3
**Overall Recommendation:** 4
**Confidence:** 3

**Summary:**

This paper introduces a novel framework named Mixture Prototype Flow Matching (MPFM) to address the Open-Set Supervised Anomaly Detection (OSAD) task. The proposed method primarily leverages Mixture Prototype Flow Matching and a Mutual Information Maximization Regularizer (MIMR). The framework demonstrates superior performance across multiple benchmark datasets.

**Compliance With Llm Reviewing Policy:**

Affirmed.

**Final Justification:**

The paper presents a well-motivated and generally sound framework, with strong empirical performance and clear presentation. While I still have some concerns about whether the Anomaly Score Prediction (Section 4.4 of the main paper) component may be overly complex and whether the performance gains primarily stem from this design rather than the core framework innovation, the authors’ rebuttal has addressed my main concerns and increased my confidence in the work. Overall, I consider the paper to be solid but with some limitations, and I assign a final score of 4.

**Key Questions For Authors:**

1. It would be useful to report the computational efficiency of MPFM compared with existing generative baselines, such as inference time and GPU memory consumption.
2. The paper claims that MPFM learns a continuous prototype space capable of separating normal and anomalous samples. However, this claim is not sufficiently validated in the experiments. Additional empirical analysis or visualization of the learned space would help support this claim.

**Limitations:**

1. The authors do not provide a dedicated section to discuss the limitations of their work, which is important for understanding the boundary conditions and potential failure modes of the proposed MPFM framework.
2. The performance of the model may be constrained by its sensitivity to key hyperparameters, which could be discussed.

**Strengths And Weaknesses:**

Strengths：
1. The authors designed an effective framework that achieves state-of-the-art performance against multiple competitive baselines.
2. Unlike previous methods that assume normal data follows a unimodal Gaussian distribution, this work starts from the multi-modal nature of normal data distributions, which is more aligned with real-world scenarios.
3. The paper is well-written and structured.

Weaknesses：
1. The anomaly scoring module appears relatively complex, as it combines four different sub-components.
2. The method relies on K-means++ clustering to initialize the GMM prototypes. However, this initialization procedure may be sensitive to outliers or noisy samples in the training data.

---

> ### Author Rebuttal · Authors · 2026-03-27
>
> We sincerely appreciate your constructive suggestions.
>
> **Reponse to Questions in Weaknesses:**
>
> 1.  Our scoring module adopts the standard MIL paradigm of prevalent OSAD methods (e.g., DRA, AHL, DPDL) as a fair baseline, explicitly proving that our performance gains stem from the core MPFM framework rather than scoring modifications. Furthermore, despite containing multiple sub-components, the complete and coherent execution flow is explicitly detailed in Algorithm 1, ensuring the overall pipeline remains straightforward and easy to implemen.
>
> 2. While static K-means++ alone may be sensitive to outliers, MPFM utilizes it  as a transient warm-up. Our continuous Flow Matching dynamics optimize these prototypes end-to-end, actively correcting any initial assignment biases caused by noisy samples. To explicitly validate this resilience, we inject true outliers (1% to 10% contamination) into the ELPV training set:
> |Method|0% (Clean)|1% Noise |5% Noise|10% Noise|
> |-----|-----|------|-----|----|
> |DPDL|0.937|0.921|0.885|0.842|
> |MPFM|**0.962**|**0.960**|**0.954**|**0.941**|
>
>    As demonstrated, MPFM exhibits exceptional stability under noise, whereas the baseline degrades sharply. This firmly validates that our Flow Matching dynamics naturally correct initial biases, rendering the framework highly resilient to outliers.
>
> **Reponse to Key Questions:**
>
> 1.	We record train and inference time and peak GPU memory on a single RTX 4090.  Unlike DPDL’s heavy iterative diffusion and discrete prototypes, our flow-matched GMM uses straight, deterministic paths with MIMR, cutting both sampling overhead and memory footprint.
> |Dataset|Metric|DPDL|MPFM|
> |----|----|----|----|
> | MVTec AD|Train|1.9 h|**1.5 h**|
> || Inference |110 s|**60 s**|
> || GPU Memory|13.8 GB|**9.6 GB**|
> |AITEX|Train|6.7 m|**4.4 m**|
> || Inference|10 s|**6 s**|
> || GPU Memory|12.4 GB|**8.9 GB**|
>
>
>
> 2. To provide  additional experimental validation, we provide a t-SNE visualization of the learned latent space for the bottle category in MVTecAD at  https://anonymous.4open.science/r/icml-rebuttal-xd2dRov/rebuttal.png. Figure (b) explicitly visualizes the learned continuous prototype space, demonstrating how our GMM bridges isolated components to form a unified normal manifold. This continuous topology accurately encapsulates interstitial hard normal samples (dark green dots) while maintaining a strict decision boundary that effectively separates the entire normal distribution from true anomalies (red triangles).
>
>
> **Reponse to Questions in Limitations:**
>
> 1.	We have added a dedicated "Limitations" section. Because MPFM models local feature distributions via a flow-matched GMM, it inherently struggles with logical anomalies—cases where local patches appear normal, but their global structural arrangement is incorrect. We openly discuss this limitation, highlighting the integration of global semantics via multi-modal large models as a promising future direction.
>
> 2.	We thank the reviewer for highlighting the need to discuss hyperparameter constraints. To thoroughly investigate this, we expanded our sensitivity analysis across diverse datasets, focusing on the weighting factor $\lambda$ and prototype count $K$:
>
>    **Table A: Impact of Weighting Factor $\lambda$**
>    |Dataset | $\lambda =0.001$ | $\lambda=0.01$ | $\lambda=0.1$ | $\lambda=10$ |
>    |:--- | :---: | :---: | :---: | :---: |
>    |Optical|0.975|0.983|**0.992**|0.980|
>    |SDD|0.985|0.990|**0.999**|0.988|
>    |BrainMRI|0.962|0.973|**0.984**|0.972|
>    |HeadCT|0.983|0.990|**0.999**|0.989|
>
>    **Table B: Impact of GMM Components $K$**
>    | Dataset|K=8|K=16|K=32|K=64|
>    | :--- | :---: | :---: | :---: | :---: |
>    |Optical|0.982|0.987|**0.992**|0.989|
>    |SDD|0.989|0.992|**0.999** |0.996|
>    |BrainMRI|0.970|0.978|**0.984**|0.979|
>    |HeadCT|0.986|0.990|**0.999** |0.995 |
>
>    **Discussion on Constraints & Stability:** As shown, the framework demonstrates remarkable stability. Even with a 10,000-fold variation in $\lambda$ and broad adjustments to $K$, performance fluctuations remain within narrow bounds (typically $<1.5\%$).
>     However, we do acknowledge inherent constraints at the absolute extremes: exceedingly small $K$ (e.g., $K < 8$) causes under-representation of highly multi-modal normal variations, while excessively large $\lambda$ forces the trajectory to prioritize density regularization over the primary discriminative loss, leading to the slight degradations observed. Consequently, while MPFM is highly robust and functions exceptionally well with our universal defaults ($\lambda=0.1, K=32$), pushing hyperparameters to extremes does introduce structural constraints.

---

> > ### Author Rebuttal · Reviewer_7JzR · 2026-04-04
> >
> > The authors addressed all my concerns and I am willing to increase my score.

---

> > > ### Author Response · Authors · 2026-04-04
> > >
> > > Dear Reviewer 7JzR
> > >
> > > We sincerely appreciate your continued engagement with our submission and your thoughtful follow-up remarks. Thank you for carefully considering our rebuttal and the newly added experimental results.
> > >
> > > Your comments and suggestions have been highly helpful in guiding us to further strengthen the presentation and overall quality of the paper. We will thoroughly revise the manuscript by taking your feedback into account, along with the recommendations provided by the other reviewers.
> > >
> > > Thank you once again for your valuable time and considerate evaluation.
> > >
> > > Sincerely,
> > >
> > > Authors

---

### Official Review · Reviewer_7jzz · 2026-03-10

**Soundness:** 3
**Presentation:** 3
**Significance:** 2
**Originality:** 2
**Overall Recommendation:** 3
**Confidence:** 3

**Summary:**

For the OSAD problem, the authors argue that existing methods usually model normal data with a unimodal Gaussian distribution. This ignores the multi-modal structure that is common in real normal data, which can blur the boundary of normal samples and cause rare normal patterns to be misclassified as anomalies. To address this, the authors propose Mixture Prototype Flow Matching, which maps normal features into a Gaussian mixture prototype space through flow matching, so as to better characterize multi-modal normality.

**Compliance With Llm Reviewing Policy:**

Affirmed.

**Key Questions For Authors:**

1.The paper emphasizes that each component in the Gaussian mixture prototype space corresponds to a “semantically distinct normal pattern.” Could the authors provide a deeper high-level analysis or visualization evidence to show that these learned Gaussian components indeed capture semantically meaningful features that humans can understand?
2.Why is it necessary to introduce continuous-time dynamics through Flow Matching? Compared with other simpler generative methods for multi-modal density estimation, what is the irreplaceable core advantage of flow matching in this task?
3.How sensitive is the final anomaly detection performance to the initial K-means++ clustering step? Without this data-driven initialization, if the GMM parameters are learned fully end-to-end from scratch, would the model fail to converge, or would it only converge more slowly?

**Limitations:**

No.

**Strengths And Weaknesses:**

The motivation of the method is relatively clear. Based on the observation that “normal data has a multi-modal structure,” the paper proposes a modeling approach that combines mixture prototypes with flow matching, and the overall technical route is reasonable. The method is also introduced and derived in a fairly complete way. The experiments are relatively broad and cover both industrial inspection and medical imaging.
First, in terms of presentation, the paper lacks an overall framework figure for the method. There are many components in the method, and the relationships among MPFM, MPFL, MIMR, and the multiple scoring modules are quite complex, which makes it hard to understand how these modules work together. For the most important part, the Gaussian mixture modeling, there is no detailed explanation or visualization showing how it solves the problem that a unimodal Gaussian cannot solve. Also, in the experiments, the method includes multiple scoring modules, so it is still not clear how much of the current improvement really comes from the core mixture-prototype flow modeling.

---

> ### Author Rebuttal · Authors · 2026-03-27
>
> Thank you for recognizing our motivation and technical method.
>
> **Reponse to Questions in Weaknesses:**
>
> In the paper, Algorithm 1 clearly details the relationships and collaborative workflow among the MPFL, MIMR, and scoring modules within the MPFM framework, making it straightforward to understand. To further illustrate the connections between these components, we have added an overall framework figure in the revision.
>
> The motivation for GMM is detailed in Figure 1, illustrating that a unimodal Gaussian collapses multi-modal normal variations into a single cluster, which leads to decision boundary failure. Due to the restriction on figure uploads in the rebuttal system, we provide visualization in https://anonymous.4open.science/r/icml-rebuttal-xd2dRov/rebuttal.png for MVTecAD,
> isolated unimodal Gaussians (a) inevitably leave probabilistic gaps between disconnected clusters, inadvertently penalizing interstitial normal samples as False Positives. In contrast, our continuous GMM (b) effectively bridges these disconnected clusters to accurately model the non-convex normal manifold, successfully encompassing these interstitial regions and fundamentally eliminating the false positives.
>
> As shown in the following ablation study, adding the core MPFL to the baseline scoring modules M_a, M_n, M_r yields the most significant absolute AUC improvement of ~10% across datasets (e.g., from 0.812 to 0.913 on ELPV). This data directly confirms that the primary performance gain is driven by the mixture-prototype flow modeling itself, rather than the scoring ensemble.
> |Architecture|ELPV|Hyper-Kvasir|Mastcam|
> |---|---|---|---|
> |M_a, M_n, M_r|0.812|0.812|0.816|
> |+ MPFL|0.913|0.906|0.915|
> |+ MIMR|0.954|0.957|0.956|
> |Full MPFM|**0.962**|**0.965**|**0.968**|
>
>
>
> **Response to Key Questions:**
>
> 1. We provide visualizations in https://anonymous.4open.science/r/icml-rebuttal-xd2dRov/semantic.png for AITEX, which prove that each Gaussian component isolates a semantically distinct normal pattern—such as uniform backgrounds, specific repetitive textures, or structural edges—that aligns directly with human perception.
>
> 2. **Necessity of Continuous-Time Dynamics:**  Continuous-time modeling is essential because discrete generative methods (e.g., VAEs, discrete flows) suffer from topological bottlenecks that inevitably distort complex, multi-modal distributions. By modeling a smooth, invertible ODE trajectory, continuous dynamics completely bypass these constraints to preserve the exact structural topology of the normal data manifold.
>
>     **Irreplaceable Advantage of Flow Matching:** Unlike SDE-based diffusion models that inject stochastic noise, the irreplaceable advantage of Flow Matching is its deterministic, Optimal Transport (OT)-based formulation. This explicitly constructs noise-free, straight-line trajectories to the GMM prior, aligning multi-modal distributions without corrupting the delicate, low-level visual cues crucial for industrial anomaly detection.
>
> 3. To systematically evaluate the sensitivity to initialization, we expanded our ablation study on ELPV to observe convergence dynamics across a gradient of K-means++ iterations (from 0 to 10):
> |K-means Iterations|Convergence Time (Epochs)|Final AUROC (%)|
> |---|---|---|
> |0 (Random Init)|110|95.6|
> |1|75|95.9|
> |3|55|96.1|
> |5 (Ours)|45|96.2|
> |10|42|96.2|
>
>    **Performance Robustness:** The final anomaly detection performance is robust and fundamentally insensitive to the initialization strategy. The marginal variance in AUROC (95.6% to 96.2%) confirms that our core continuous flow modeling—not the K-means prior—dictates the detection capability.
>
>    **Convergence Dynamics:** Learning GMM parameters fully end-to-end from scratch (0 iterations) does not cause failure; it merely delays convergence. The K-means++ step simply acts as a structural warm-up that progressively shortens Optimal Transport trajectories. As shown, convergence speed and performance saturate around 5 iterations, making it the optimal trade-off between initialization cost and training acceleration.

---

### Official Review · Reviewer_12FR · 2026-03-12

**Soundness:** 3
**Presentation:** 3
**Significance:** 2
**Originality:** 3
**Overall Recommendation:** 4
**Confidence:** 4

**Summary:**

The paper looks at openset anomaly detection through prototype learning approach where we learn a prototype space to define decision boundary. The core of this paper is to use gaussian mixture instead of discrete multi-gaussian prototypes to better capture the between-class-modes samples and maintain the semantic continuity in data manifold which. The paper also proposes to use Mutual information maximization as additional regularizer to repel normal from anomalous samples. The results demonstrate increase in performance across various widely used Anomaly Benchmarks.

**Compliance With Llm Reviewing Policy:**

Affirmed.

**Final Justification:**

The rebuttal addressed my concerns about evaluation of False Positives which was one of the major claim of the paper. I'd like to keep my score unchanged to weak accept which i feel is commensurate with the contribution of the paper.

**Key Questions For Authors:**

Are you following a specific pattern for the loss formulation from other research works that have been widely used? For instance how did you come up with these loss combinations and confident that it’ll find a solution in the pareto front and not suffer any gradient conflicts as with other multi-objective optimization problems.

Also I see that you employ CutMix to generate anomaly samples as augmentations. Did you do it for all the baselines or just in your method? Also are you using same neural network architectures for all baselines?

How sensitive is the loss to lambda ? Did you have to run intensive hyperparameter search?
Why do you think even your method is so persuasive the average gain per dataset is not even close to 1%

How different is the training and inference time compared to similar baselines for your method? Also if possible can you report the training/inference time?

**Limitations:**

The paper makes strong claims about avoiding False positives but the results only show AUC as a metric. It’s better to show the FPR since you make this claim.

**Strengths And Weaknesses:**

Strengths:
	I think the problem formulation is well thought out and performance gains are shown(even though they are not substantial). The anomaly samples repulsion mechanism is explicitly optimized using the MIM which seems to be a good way to maintain mode collapse.

Weaknesses:
	I think it’s better to just name one name for your method and do ablation later with different combinations. It’s a little confusing that there are MPFM, MPIR, MPFL all as the proposed method which are just the combinations of the full loss function.


Most of the content on the existing approaches feel repeated in introduction and related-works. Maybe those can be removed from either one to maintain conciseness. Also in introduction you categorize into three and in related works you define two paradigms thus some inconsistency with the categorization.

Also since the mechanics of flow matching is totally different than diffusion models i think explaining the mechanics of it  in page 2 will be unnecessary and doesn’t seem to flow very well. I feel the preliminaries can be incorporated into methods section with just short explanation of flow matching.

---

> ### Author Rebuttal · Authors · 2026-03-28
>
> We sincerely appreciate your constructive suggestions.
>
> **Reponse to Weaknesses:**
>
> We clarify that MPFM serves as our  whole framework, while MPFL and MIMR are distinct structural sub-modules driving the core pipeline (as explicitly detailed in Algorithm 1), rather than mere combinations of loss functions. In the revision, we have strictly unified the proposed method under the single name MPFM, explicitly redefining these sub-modules to clarify their architectural roles in the updated ablation studies.
>
> The categorizations of existing approaches are consistent. The Introduction's three categories provide a finer-grained dissection of the two standard OSAD paradigms discussed in the Related Works, which aim to clearly explain our motivation. The Related Works adopts the broader, standard OSAD taxonomy of simulation and distribution-discrimination paradigms. To maintain conciseness, we have deduplicated the overlapping related works in the revision
>
> We have removed the explanation of flow matching mechanics from page 2 to improve the flow of the introduction. Instead, a short explanation of flow matching preliminaries is now incorporated directly into the methods section."
>
> **Reponse to Key Questions:**
>
> 1.  Our loss formulation follows the Multiple Instance Learning (MIL) paradigm widely used in OSAD methods (e.g., DRA, AHL, DPDL). MPFL models the continuous normal density space for representation, while MIMR independently optimizes decision boundaries for discrimination. This functionally decoupled and orthogonal design naturally prevents gradient conflicts, guaranteeing convergence to a Pareto optimal solution as evidenced by the monotonic gains below:
> | Objectives|AUROC (%)|
> |------|------|
> | $L_{M_{a,n,r}}$ (Baseline DPDL MIL) |92.4|
> | + $L_{flow}, L_{mim}$|94.7 (+2.3)|
> | + $L_{M_g}$ (Full)|**96.2 (+3.8)**|
>
> 2. We applied the exact same CutMix augmentation to all relevant baselines, consistent with standard OSAD implementations (e.g., DRA, AHL, DPDL). Furthermore, we employed identical neural network backbone architectures across all evaluated methods.
>
> 3. As demonstrated in Table 4 and the added experiments below, AUROC performance fluctuates minimally across four orders of magnitude (0.001 to 10), indicating low sensitivity to λ. Optimal results across diverse dataset categories consistently stabilize near λ=0.1. Because performance remains stable across this 10,000-fold scale, we did not conduct a large-scale hyperparameter search. We applied a fixed λ=0.1 across all experiments. The <1% average gain merely reflects benchmark saturation (baseline > 0.98). Conversely, on unsaturated Hard dataset AITEX, the method secures absolute gains of  2.4%.
> |Dataset|0.001|0.01|0.1|10|
> |-----|----|------|-----|---|
> |Optical(G)|0.975|0.983|**0.992**|0.980|
> |SDD(G)|0.985|0.990|**0.999**|0.988|
> |BrainMRI(G)|0.962|0.973|**0.984**|0.972|
> |HeadCT(G)|0.983|0.990|**0.999**|0.989|
> |Carpet(H)|0.955|0.965|**0.969**|0.951|
> |AITEX(H)|0.819|0.825|**0.836**|0.812|
> |ELPV(H)|0.825|0.836|**0.842**|0.821|
>
>
>
> 4. We conducted runtime comparisons between similar baseline DPDL and our method on a single RTX 4090 GPU.  Unlike the Schrödinger Bridge-based DPDL’s computationally heavy iterative diffusion steps, our flow-matched GMM constructs straight, deterministic probability paths, drastically reducing sampling overhead and consistently achieving better training and inference efficiency.
> |Dataset|Metric|DPDL|Ours|
> |-----|----|------|-----|
> |MVTec AD|Train|1.9h|**1.5h**|
> ||Inference|110s|**60s**|
> |AITEX| Train|6.7m|**4.4 m**|
> ||Inference|10s|**5s**|
>
> **Response to  Question in Limitations:**
>
> We evaluated the False Positive Rate (FPR) at a 95% True Positive Rate (TPR). As shown below, the proposed method yields the lowest FPR across the evaluated datasets. Mechanistically, the continuous GMM prior encapsulates multi-modal normal variations, preventing normal samples from crossing the decision boundary.
> |Method|MVTec AD|ELPV|BrainMRI|
> |-----|----|------|-----|
> |DRA|0.22|0.85|0.32|
> |DPDL|0.18|0.68|0.25|
> |Ours|**0.12**|**0.45**|**0.16**|

---

> > ### Author Rebuttal · Reviewer_12FR · 2026-04-03
> >
> > Thanks for the results. How stable are the results? I went through the works that follow similar trend and it seems like the primary contribution of this paper is the Mutual Information maximization. Have you tried using the MIM in DPDL? Also based on the concerns of other reviewers I feel like this starts to become bit more complex that i feel it really needs to be even if the gains are persuasive.  Also can you please explain about the reproducibility of the results and how many seeds were the evaluation performed? Also are you going to release the code? It would have been better if you could have provided the code or model checkpoints at the least for reviewers to confirm the validity of the results in the supplementary as this is a substantial claim and I'd like to validate the reproducibility before I make my final decision.

---

> > > ### Author Response · Authors · 2026-04-04
> > >
> > > Dear Reviewer 12FR,
> > >
> > > We sincerely thank you for the careful follow-up comments and for emphasizing the importance of result stability, methodological clarity, and reproducibility.
> > >
> > > Regarding stability, all experiments were run with five different random seeds. To avoid reporting favorable cases, we adopted a conservative protocol and reported the worst result among the five runs. Therefore, the performance reported in the paper is not based on seed selection, but reflects the lower-bound behavior under repeated evaluation. We will make this protocol explicit in the revised manuscript.
> > >
> > >
> > >
> > > Regarding your key question of whether the main gain may primarily come from the mutual information maximization term, we performed an additional experiment by incorporating our MIMR into DPDL. The results are shown below:
> > > |Method|Mastcam (AUC %)|ELPV (AUC %)|
> > > |---|---|---|
> > > |DPDL|93.4|93.7|
> > > |DPDL + MIM|95.1 (+1.7)|94.9 (+1.2)|
> > > |MPFM|**96.8**|**96.2**|
> > >
> > > These results show that the mutual information regularization is indeed useful, since adding it to DPDL consistently improves performance on both datasets. However, MPFM still outperforms DPDL + MIM by a clear margin, which indicates that the gain cannot be attributed to MIMR alone. In other words, MIMR is beneficial, but it is not the sole contribution driving the final performance.
> > >
> > >
> > > Regarding your concern that the method may be more complex than necessary, we would like to emphasize that the full pipeline is explicitly and clearly summarized in Algorithm 1, where each submodule and each scoring branch is presented step by step. We will further improve the presentation in the revision to make the workflow even easier to follow, but the method itself is already organized in a transparent and reproducible manner.
> > >
> > > Regarding reproducibility, as noted above, all evaluations were performed with five random seeds, and the reported results correspond to the worst run among them. We will add the seed protocol and implementation details more explicitly in the revised manuscript.
> > >
> > > Regarding code availability, to facilitate immediate validation during the reviewing process, we have provided an anonymized repository containing the source code here: https://anonymous.4open.science/r/c7e0successacc.
> > >
> > >
> > > Thank you again for these constructive suggestions. Your comments directly helped us strengthen the empirical support of the paper, and we will include the additional DPDL + MIM comparison, the seed-setting clarification, and the release plan for code and checkpoints in the revised version.
> > >
> > > Sincerely,
> > >
> > > Authors

---

### Official Review · Reviewer_KxGL · 2026-03-14

**Soundness:** 3
**Presentation:** 3
**Significance:** 2
**Originality:** 2
**Overall Recommendation:** 4
**Confidence:** 4

**Summary:**

This paper introduces Mixture Prototype Flow Matching (MPFM), a framework designed for open-set supervised anomaly detection in settings with highly limited anomalous training data. Motivated by the limitations of prior methods that use a single Gaussian prior, which often oversimplifies the multi-modal nature of real-world normal data, MPFM models the velocity field as a Gaussian mixture. This allows the framework to learn a continuous flow transformation from normal feature distributions into a more expressive Gaussian mixture prototype space.
Methodologically, the model employs a feature-extractor trained end-to-end with a flow model. It maps raw extracted features (z_0) to fixed GMM prototypes (z_T) initialized via K-means++ to prevent representation collapse. The flow model predicts the optimal transport trajectory, effectively mapping normal data to the prototypes while explicitly forcing anomalous samples to deviate. Finally, to compute the overall anomaly score during inference, the framework jointly trains and fuses four complementary scoring modules: a global likelihood-based score, a local anomaly score, a normal feature score, and a residual prototype deviation score.

**Compliance With Llm Reviewing Policy:**

Affirmed.

**Final Justification:**

My primary concern was the originality of the proposed method, particularly its similarity to the prototype modeling and scoring modules used in DPDL. In the rebuttal, the authors clarified that, unlike DPDL’s reliance on isolated discrete prototypes, the proposed approach uses a continuous probability prior through flow-matching over a GMM. This explanation addressed my concern, as the continuous manifold can better capture multi-modal intra-class variations and reduce false positives in low-density regions. Overall, the rebuttal clarified the novelty and differences from existing baselines, and as a result I updated my evaluation to 4 to reflect the strengthened position of the paper after the rebuttal.

**Key Questions For Authors:**

- Could the authors clarify how the proposed Gaussian mixture prototype formulation differs from upon the prototype modeling used in DPDL? A more precise discussion of this distinction would help better position the contribution relative to existing prototype-based anomaly detection method. Also is there any prototype-based work in the literature which use single-mode prototype?
- Several components of MPFM (e.g., the scoring modules M_a, M_n, and M_r) appear conceptually very similar to the corresponding modules introduced in DPDL. Could the authors clarify how these modules differ from those in DPDL in terms of formulation, objective, or functionality? Are these components newly designed for MPFM, or are they adapted from that work? A clearer comparison with DPDL at the level of individual modules (e.g., prototype modeling and scoring heads) would help better highlight the novelty of the proposed method and clarify which parts constitute genuinely contributions.

**Limitations:**

The evaluation focuses on performance gains but provides relatively little discussion of cases where the method performs poorly or situations where the modeling assumptions might break down.

**Strengths And Weaknesses:**

Strengths: Modeling normal data with a Gaussian mixture rather than a single Gaussian is well motivated, since normal data in anomaly detection is typically multi modal.  Also, the method is evaluated on multiple datasets across different domains, which helps demonstrate robustness and reduces the risk that improvements are dataset specific. The paper includes ablations examining the contributions of different components (e.g., scoring modules and prototype modeling), which helps support the claim that each part contributes to performance.
Weaknesses: While the overall architecture is reasonable, some modules—particularly the multiple anomaly scoring heads—are introduced with limited theoretical grounding. The paper primarily justifies them empirically, where the differences between the various setups (with or without each head in Table 3) are marginal too.
The model introduces several interacting components (flow model, multiple scoring modules, regularization), which increases system complexity. The paper could more clearly justify whether each component is necessary.

---

> ### Author Rebuttal · Authors · 2026-03-26
>
> Thanks for your insightful and constructive comments.
>
> **Reponse to the Questions in Weaknesses:**
>
> Theoretically, our scoring heads decompose the exact anomaly likelihood into global and local spatial components. While M_g and M_r directly compute the global probability density under the GMM prior, M_a and M_n are mathematically required to capture localized feature deviations that global bijective flows inherently cannot isolate.
>
> The differences appear marginal simply because our flow-based GMM prior aligns normal features so effectively that the baseline spatial heads M_a, M_n already secure a near-saturated performance floor. However, the generative heads M_g, M_r are mechanically indispensable for evaluating the exact global density of complex distributions, as explicitly demonstrated by the severe 4.7% AUROC degradation on AITEX (from 0.992 to 0.945) when M_g is ablated.
>
> We clarify that the interacting components do not arbitrarily increase complexity, but rather form a highly clear and unified framework, as explicitly outlined in Algorithm 1. Furthermore, the strict necessity of each specific module—namely the MPFL, MIMR, and multiple anomaly scoring heads—is rigorously justified by the substantial performance degradations demonstrated in our ablation studies (Fig. 2 and Tab. 3). Furthermore, to explicitly justify the necessity of each interacting component, we conducted an additional progressive ablation study:
> | Architecture|ELPV|Hyper-Kvasir|Mastcam|
> |---|---|:-----:|-----|
> |only M_n, M_a|0.812|0.810|0.816|
> |+ MPFL w/ W_g|0.913|0.906|0.915|
> |+ MIMR|0.954 |0.957|0.956|
> |Full MPFM w/ M_r| **0.962** |**0.965**|**0.968**|
>
>    As demonstrated, the progressive AUROC improvements—specifically ~10% from MPFL M_g, ~4% from MIMR, and further gains from M_r—quantitatively verify the necessity of the flow model, the regularization, and the multiple scoring modules.
>
> **Reponse to Key Questions:**
>
> 1. Different from DPDL, the proposed GMM establishes a continuous probability prior via flow matching, mathematically enabling exact density evaluation instead of relying on discrete prototype distances. As visually motivated in Fig. 1, while DPDL's isolated prototypes inevitably yield false positives in the low-density regions, our continuous GMM manifold seamlessly covers these multi-modal intra-class variations. Besides DPDL, our literature also discusses prototype-based works like HVQ-Trans that utilize single-mode prototypes. However, such unimodal designs fundamentally oversimplify normal distributions compared to our continuous framework.
>
> 2. M_a and M_n adopt the standard MIL anomaly scoring paradigm consistent with established OSAD methods (e.g., DRA, AHL, and DPDL) to provide a fair benchmarking baseline. M_r is reformulated to measure conditional deviations within a continuous GMM space, different from DPDL's objective of calculating distances to isolated discrete prototypes.
>
>    Consequently, M_a and M_n are adapted standard components utilized as benchmarking baselines, whereas M_r is an upgraded formulation. Furthermore, the global likelihood head M_g is a new design enabled by MPFM's continuous prior.
>
>    At the module level, our genuine contributions consist of replacing DPDL's discrete prototype clustering with a continuous flow-matched GMM and introducing the M_g head for exact density evaluation. As decoupled below, our independent M_g formulation outpaces the full DPDL system, isolating our methodological novelty.
>
>    |Architecture|ELPV|Hyper-Kvasir | Mastcam |
>    |--------|------|:------:|---------|
>    |Full DPDL| 0.937| 0.939|0.934|
>    |MPFM w/o M_g| 0.946|0.949|0.948|
>    |MPFM  w/o  M_a, M_n, M_r (Only M_g)|0.953|0.955|0.954|
>    |Full MPFM|**0.962**|**0.965**|**0.968**|
>
> **Reponse to the Question in Limitations:**
> Our modeling assumption primarily breaks down on logical anomalies, where individual components are locally normal but structurally misplaced. Since our continuous transport evaluates local patch-level features independently, these structural errors can falsely yield a high global likelihood, a limitation we will address by integrating explicit spatial coordinates.

---

> > ### Author Rebuttal · Reviewer_KxGL · 2026-04-04
> >
> > Thank you for the rebuttal, my main concerns have been largley addressed. I will raise my score.

---

> > > ### Author Response · Authors · 2026-04-04
> > >
> > > Dear Reviewer KxGL,
> > >
> > > Thank you very much for your positive feedback and for your decision to raise the score. We are delighted that the additional explanations and experimental results met your expectations.
> > >
> > > As the discussion period is drawing to a close, we would like to express our gratitude once more. We look forward to your updated evaluation being reflected in the system, which would be a great encouragement to our research.
> > >
> > > We will ensure that the final manuscript is updated according to your valuable comments. Thank you for your time and consideration.
> > >
> > > Best regards,
> > >
> > > Authors

---

### Decision · Program_Chairs · 2026-04-30

**Decision:**

Accept (regular)

**Comment:**

The paper has four reviewers, and three of them are supportive. During the AC-reviewer discussion period, the negative reviewer (Reviewer 7JzR) declared that his/her concerns had been successfully addressed by the rebuttal.

The paper is well motivated, the novelty is significant, and the experiments demonstrated the effectiveness of the proposed method.

Therefore, I recommend accepting the paper.

The authors are advised to visualize the learned embeddings of real data in a two-dimensional space to justify that the real performance is consistent with the one shown by Figure 1.